# DMAP: a Distributed Morphological Attention Policy for Learning to Locomote with a Changing Body

Alberto Silvio Chiappa      Alessandro Marin Vargas      Alexander Mathis

École Polytechnique Fédérale de Lausanne (EPFL), Switzerland
`[first name].[surname]@epfl.ch`

## Abstract

Biological and artificial agents need to deal with constant changes in the real world. We study this problem in four classical continuous control environments, augmented with morphological perturbations. Learning to locomote when the length and the thickness of different body parts vary is challenging, as the control policy is required to adapt to the morphology to successfully balance and advance the agent. We show that a control policy based on the proprioceptive state performs poorly with highly variable body configurations, while an (oracle) agent with access to a learned encoding of the perturbation performs significantly better. We introduce DMAP, a biologically-inspired, attention-based policy network architecture. DMAP combines independent proprioceptive processing, a distributed policy with individual controllers for each joint, and an attention mechanism, to dynamically gate sensory information from different body parts to different controllers. Despite not having access to the (hidden) morphology information, DMAP can be trained end-to-end in all the considered environments, overall matching or surpassing the performance of an oracle agent. Thus DMAP, implementing principles from biological motor control, provides a strong inductive bias for learning challenging sensorimotor tasks. Overall, our work corroborates the power of these principles in challenging locomotion tasks. The code is available at the following link: DMAP

## 1  Introduction

Animals and humans are highly adaptive to changes in the external environment and in their own body. For instance, animals can, after an early onset, locomote throughout their development despite dramatic changes in their body size and weight. Animals can also robustly deal with irregular surfaces, and load changes [1, 2, 3, 4, 5].

Here, we focus on the problem of learning to walk when the body is subject to morphological perturbations, i.e., changes in the length and thickness of body parts. Remarkably, animals can rapidly adapt to these kind of perturbations. For example, desert ants with elongated or shortened legs can immediately locomote, although misjudging the traveled distance [6]. To challenge artificial agents on such skills, we adapted four different locomotion environments (Ant, Half Cheetah, Walker, Hopper) from the OpenAI Gym [7], in which robots have to learn to walk as fast as possible. In our adaptive setting, the agent's body parameters are sampled from a morphological perturbation space at the beginning of each episode and we sought to find reinforcement learning policies that could deal with these varying bodies.

We hypothesize that principles of the sensorimotor system provide a strong inductive bias to robustly learn such a policy. Specifically, we built on three well-known principles:

36th Conference on Neural Information Processing Systems (NeurIPS 2022).

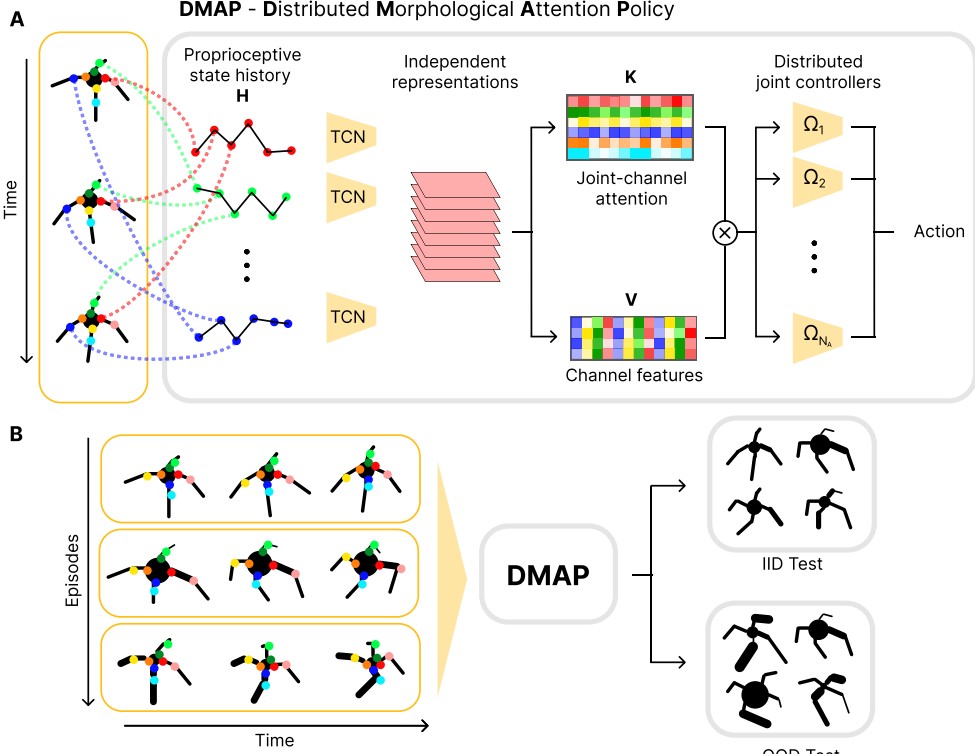

Figure 1: **A**. Overview of the architecture of DMAP. Each component of the state history is processed independently by a temporal convolutional network (with shared weights). The extracted features are transformed into the value encoding matrix and the attention weights of each joint sensor and action component. The product of these matrices results in a joint-specific morphology encoding, which is the convex combination of the value encoding vectors, weighted by their attention score. Independent fully-connected networks control a single joint, based on the current proprioceptive state and the morphology encoding **B**. DMAP is trained for several episodes with randomly sampled perturbed bodies within the training distribution. At test time, the zero-shot performance of the model is assessed on a fixed set of body configurations, sampled either from within (IID) or outside (OOD) of the training distribution.

**Independent low-level processing.** Somatosensory inputs are first processed locally (e.g., by temporal filtering) before they are integrated across body parts and represented in an topographically ordered fashion [8, 9, 10].

**Gated connectivity.** Sensory information is dynamically gated in a task dependent manner [11, 12], and integrated with action and value signals in the brain [13, 10, 14].

**Distributed control.** Different body parts are controlled in a distributed fashion [9, 10, 5].

Based on these principles, we propose the Distributed Morphological Attention Policy (DMAP) architecture (Figure 1). We show that DMAP can robustly learn to locomote with changing bodies without access to the morphological parameters. To assess the performance of DMAP, we compare it to several alternative agents: Simple, Oracle and rapid motor adaptation (RMA) [15]. Experiments reveal that an agent with access only to the proprioceptive state, consisting of its joint angles and velocities (*Simple*), fails to learn an effective locomotion strategy for a large variety of body configurations. In contrast, augmenting the observation with the current body parameters leads to a strong ceiling performance (*Oracle*). Remarkably, for all different environments, DMAP is as good as Oracle. We also compare DMAP to a powerful, recently proposed two-step training technique called rapid motor adaptation (*RMA*) [15]. This algorithm trains a temporal convolutional network to infer a context encoding from a short history of proprioceptive state, imitating the Oracle's morphology encoder. While RMA can also reach Oracle performance, we show that end-to-end training of such a

network does not succeed for more complicated environments (Ant and Walker). In contrast, DMAP's policy can be trained end-to-end *without* access to the morphological parameters and reaches or surpasses Oracle performance for all four environments (Ant, Half Cheetah, Walker, Hopper).

Overall, our work suggests that independent low-level processing, distributed control and dynamic gating mechanisms provide a strong inductive bias for sensorimotor learning. Finally, we analyze the information gating mechanisms of DMAP. The strength of gated connections reveals morphology-specific patterns, such as bilateral and back-front symmetry. The low dimensional embedding of the gating mechanism exhibits rotational dynamics, robustly across different morphologies.

## 2   Related work

**Benchmarks for zero-shot adaptation.** The setup of the experiments described in this paper can be framed as a zero-shot adaptation problem, as the agents are evaluated starting from the first moment in which they interact with an unseen context [16]. While classic RL benchmarks, such as Arcade [17] and rllab [18], assess the ability of an agent to maximize the cumulative reward in the training environment, more recent ones introduce perturbations in the dynamics [19, 20], in the observation function [21, 22] or in the reward function [23] at test time. For a more complete list, we refer to the survey by Kirk et al. about generalization in Deep Reinforcement Learning [16]. Mann et al. showed that the performance of RL agents under morphological distributional shifts correlates with the ability of internal models to generalize [24]. We took their design of environments as an inspiration for the design of morphological changes in our work.

**Context encoding and adaptation.** Algorithms which enable learning in changing environments are important for the sim-to-real transfer problem, where domain randomization represents a successful strategy to bridge the reality gap [25, 26, 27]. A notable approach to solving the zero-shot transfer problem consists in encoding the context into a hidden state, e.g., by online system identification [28], or with a recurrent neural network (RNN) [18, 29]. However, while off-policy continuous control algorithms such as TD3 [30] and SAC [31] are the current state of the art among model-free RL algorithms, both in sample efficiency and final performance [32], training a recurrent neural network off-policy presents relevant technical challenges [33].

The problem of adaptation to unseen environmental conditions is central in the Meta-RL literature [34]. Both model-based (ReBAL, GrBal [35], MOLe [36]), and model-free (PEARL [37]) algorithms prove capable to accelerate adaptation to unseen contexts. However, they require an adaptation phase in the test environment, which can last a few episodes [37] or some hours [35], seemingly suboptimal in our experimental setup (Appendix A.4). As the focus of our research is on the development of a network architecture to enable robust locomotion in a zero-shot manner, we chose to compare it to RMA, a powerful system identification approach by Kumar et al. [15]. They showed that a context encoding can be successfully inferred from a short sequence of past states and actions, and this context encoding can guide a robust policy [15].

**Distributed control.** An agent can learn to generalize to multiple body shapes by using modular policies to control each joint independently, e.g., by treating each joint [38] or group of joints [39] as a node of a graph. For example, *Nervenet* [40] achieves state-of-the-art performance on standard RL benchmarks by leveraging different policies for each joint type. However, it cannot generalize to different designs in a zero-shot manner (without fine-tuning). On the other hand, Shared Modular Policies (SMP) [41] employs the same policy for each joint, enabling an agent to deal with variable observation and action space sizes. Further solutions to handle bodies with variable connectivity graphs include transformer-based solutions, such as Amorpheus [42] and the recently proposed AnyMorph [43]. While these works study the problem of controlling bodies with a variable number of segments and connectivity graph, we focus on perturbations which affect morphological parameters, but without changing the body structure.

**Hierarchical RL** A common approach to solve complex RL tasks is to decompose them in a hierarchy of subproblems [44]. Walking with different body shapes could be interpreted as the union of a long-horizon task (locomotion) and a short-horizon one (body shape identification). Algorithms to tackle multi-horizon control problems by separating high-level behavior from motor primitives include Hierarchical Actor-Critic [45], HeLMS [46] and HiPPO [47].

# 3 A biology-inspired architecture to handle morphological perturbations

Adaptation to morphological perturbations requires an agent to rapidly identify which joints are affected and what compensatory torques are required. We propose a Distributed Morphological Attention Policy (DMAP) to address this problem (Figure 1). With DMAP, we seek to facilitate this process by promoting the formation of communication pathways from individual body parts to the specific joint controllers. The architecture is designed according to three principles of (biological) sensorimotor control (also mentioned in the introduction):

**Independent low-level processing.** Each proprioceptive channel is first processed independently, in order to obtain a channel-specific representation. This is inspired by the processing in the proprioceptive pathway [9, 10, 11].

**Gated connectivity.** An attention mechanism assigns the connection weight between the control policy of a joint and the features of each component of the proprioceptive state. This is inspired by dynamic gating mechanisms in the brain [11, 12, 13].

**Distributed control.** Independent networks control each of the agent's joints, by outputting one single action scalar (corresponding to a joint's torque). This is inspired by the architecture of the spinal cord and the motor homunculus [8, 2, 10].

We implemented those principles in the following way (Figure 1 and for more details Appendix A.1):

## 3.1 Independent processing and attention-based feature encoder

A feature encoder receives in input a history of $T$ proprioceptive states and actions $((\mathbf{s}^{(t-T)}, \mathbf{a}^{(t-T)}), ..., (\mathbf{s}^{(t-1)}, \mathbf{a}^{(t-1)}))$, which we denote as $H^{(t)} \in \mathbb{R}^{(N_S+N_A) \times T}$, $N_S$ and $N_A$ being the state and action space size, respectively. Each row $\mathbf{h}_i^{(t)} \in \mathbb{R}^T$ of $H^{(t)}$ is processed independently by the same Temporal Convolutional Network (TCN). A learned linear transformation maps each temporally-filtered input channel into the vectors $\mathbf{k}_i^{(t)} \in \mathbb{R}^{N_A}$, representing the attention of each joint towards that input channel $i$, and $\mathbf{v}_i^{(t)} \in \mathbb{R}^{N_V}$, a value encoding vector of size $N_V$ (a hyperparameter of the network). The vectors $\mathbf{k}_i^{(t)}$ and $\mathbf{v}_i^{(t)}$ from different channels are stacked to form the matrices $\tilde{K}^{(t)} \in \mathbb{R}^{(N_S+N_A) \times N_A}$ and $V^{(t)} \in \mathbb{R}^{(N_S+N_A) \times N_V}$. The matrix $K^{(t)}$ is obtained by applying a softmax operation to each column of $\tilde{K}^{(t)}$. The context encoding vectors for each joint controller are the rows $\mathbf{e}^{(t)T}$ of the matrix $E^{(t)} = K^{(t)T} V^{(t)}$, which are convex combinations of the features extracted from each sensory modality, weighted by their attention score (See diagram in Appendix A.1). This attention mechanism enables each controller to focus on the relevant morphological information carried by specific sensory channels, while ignoring the irrelevant ones. Furthermore, the attention matrix is conditioned to the recent transition history, making it adaptable both across episodes (and thus morphological states) and within a single episode.

## 3.2 Distributed joint controllers

Differently from standard policy networks, we adopt independent controllers $\Omega_i$, $i \in 1, .., N_A$ for each joint, implemented as distinct fully-connected networks. Each network outputs the element $a_i^{(t)}$ of the action vector $\mathbf{a}^{(t)}$ based on the current state $\mathbf{s}^{(t)}$, the previous action $\mathbf{a}^{(t-1)}$ and the context encoding $\mathbf{e}_i^{(t)}$. Distributed action selection is a natural way to enable the emergence of sensorimotor pathways from the sensory data of different body parts to the controller of a joint. As one controller needs to predict one single action value, we show in our experiments that smaller fully-connected networks are sufficient to successfully solve the considered tasks, limiting the complexity overhead stemming from distributed control (See hyperparameters in Appendix A.2). Qualitatively, independent policy networks can more easily adapt their behavior when a morphological perturbation affects a localized area of the body, leaving the control policy of the other networks unaltered.

# 4 Morphological perturbation environments

We consider four locomotion environments of the PyBullet physics simulator [48]: Hopper, Walker, Ant and Half-Cheetah, which are standard benchmarks for continuous control reinforcement learning

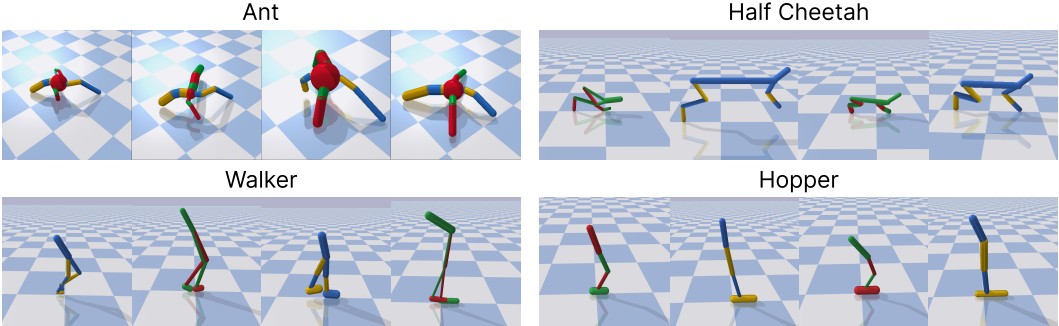

Figure 2: Example of perturbed body configurations for the four agents, sampled at the $\sigma = 0.5$ perturbation level. The segments composing the body vary in thickness and length.

algorithms [49, 32]. To test the ability of an agent to learn locomotion with a changing body, we introduce a set of morphological perturbations for each of them. The perturbations consist in modifying the size and the length of one or more body parts (Appendix A.3). Such perturbations strongly deteriorate the performance of an agent trained in the unperturbed environment [24]. The parameter $\sigma$ controls the size of the perturbation space $C_\sigma$, limiting the intensity of the perturbation between $-\sigma$ and $\sigma$. The perturbation is defined as the relative deviation of a morphological parameter from its original value. For example, a perturbation of 0.5 applied to a leg thickness will make it up to 1.5 times thicker. This construction results in widely varying body shapes for each of the four environments (Figure 2).

As in the unperturbed environments [32], the training is organized in episodes of at most 1000 time steps. At the beginning of each episode, bodily parameters are sampled uniformly at random in the space of admissible perturbations (and fixed throughout this episode). This setup defines a Contextual Markov Decision Process [16] $\mathcal{M} = \langle \mathcal{S}, \mathcal{A}, \mathcal{O}, \mathcal{T}, \mathcal{R}, \gamma, C_\sigma \rangle$. $\mathcal{M}$ is determined by the state space $\mathcal{S}$, the observation function $\mathcal{O}$, the action space $\mathcal{A}$, the transition dynamics $\mathcal{T}$, the reward function $\mathcal{R}$, the discount factor $\gamma$ and the context $C_\sigma$. The observation only includes proprioceptive information, in the form of angular position and velocity of the joints, besides other non-visual variables (orientation of the body, contact forces with the floor; see Appendix A.3). The observation space $\mathcal{S}$ does not contain explicit information about the current context, as identical observations can come from bodies with different parameters. On the other hand, the context $c_\sigma \in C_\sigma$ influences the transition dynamics $p(\mathbf{s}_{t+1} | \mathbf{s}_t, \mathbf{a}_t, c_\sigma)$, modifying the effect of an action depending on the body configuration. This condition of partial observability makes the locomotion problem considerably harder. However, confirming the intuition that contextual parameters influence the transition dynamics, we show that an agent can adapt by extracting information from the movement history.

Table 1: The five policy networks analyzed in the experiments. MLP: multi-layer perceptron. TCN: temporal-convolutional network. 2-step: imitation of the Oracle morphology encoder. P: proprioceptive state. M: morphology information. H: transition history.

|          | Simple     | Oracle     | RMA    | TCN        | DMAP            |
|----------|------------|------------|--------|------------|-----------------|
| Encoder  | -          | MLP        | TCN    | TCN        | Attention       |
| Policy   | MLP        | MLP        | MLP    | MLP        | Distributed MLP |
| Input    | P          | P + M      | P + H  | P + H      | P + H           |
| Training | End-to-end | End-to-end | 2-step | End-to-end | End-to-end      |

## 5   Experiments

We trained all the agents with Soft Actor Critic (SAC) [31], using the open source library RLLib [50]. The experiments have been run on a local CPU cluster, for a total of approx. 100 000 cpu-hours. We used similar hyperparameters to Raffin et al. [32], which provide state-of-the-art results in the base versions of the four PyBullet environments considered in our work (Appendix A.3). We ran every experiment for 5 random seeds. We found that the training of the critic network benefits slightly from augmenting the input observation with the current morphological perturbation. Therefore, we

included it for all policy architectures, except for the Simple agent. This addition does not pose any limitation to the deployment of the proposed algorithms, as the policy network alone is necessary at that stage.

The experiments involve five different policy networks (Simple, Oracle, RMA, TCN, DMAP), which differ in architecture, required input data and training procedure (Table 1). During training, the morphology parameters are randomly sampled from a perturbation space, defined by the parameter $\sigma$, at the beginning of each episode. An episode lasts 1000 time steps, unless an early termination condition is met (in case the agent falls). In this time, the agent has to learn to adapt to its unseen body configuration and run as fast as possible.

We evaluate the performance of a trained agent on its ability to adapt in a zero-shot manner to an unseen body morphology. To do so, we use a test set of 100 body configurations for each perturbation intensity, which were not part of the body configurations used during the training phase. The agent is tested on a single episode per test morphology, in which it only observes the proprioceptive states (without any information about the body state - except for the Oracle). During the testing phase, the agent does not observe the rewards it collects. We apply this testing protocol to the performance evaluation of all the trained policies and all the environments and perturbation levels. To evaluate the agent's ability to adapt to an unseen morphology, we consider the reward it accumulates in the complete test episode (without changing the policy; zero-shot setting). We do so for all the body configurations in the test set. Table 2 and the tables in Appendix A.4 list the zero-shot performance of all the trained agents in different test environments. The rewards in bold are significantly higher according to a paired t-test (p-value < 0.001). We denote the tests as IID if the 100 body configurations were extracted from the morphological space used during the training ($\sigma_{train} \geq \sigma_{test}$), otherwise we denote them as OOD ($\sigma_{train} < \sigma_{test}$).

## 5.1 A morphology encoding boosts learning in strongly perturbed environments

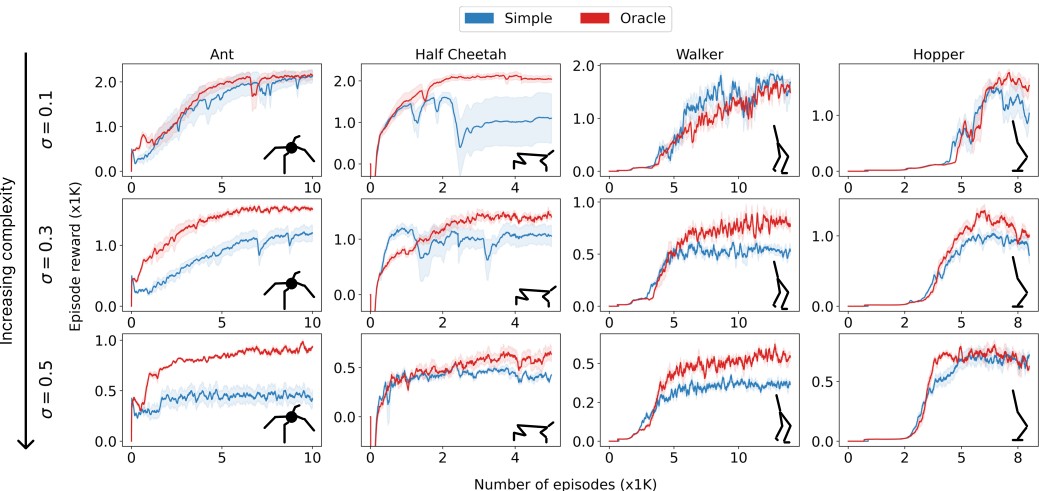

Figure 3: Learning curves of Simple and Oracle (mean ± SEM across 5 random seeds). While the Simple performs competitively in low perturbation environments ($\sigma = 0.1$), Oracle learns better at higher perturbation levels.

To assess whether an agent can benefit from an encoding of its morphology when learning a locomotion policy, we compare the performance achieved when providing only the proprioceptive state (Simple) and when augmenting it with the perturbation parameters (Oracle). Specifically, this Oracle follows the architecture of Kumar et al. [15]. It extracts a morphology encoding from the raw perturbation vector and concatenates it to the observation vector, thus augmenting the input of the policy network and breaking the partial observability of the environment. We found that, while in mildly perturbed environments Simple performs competitively with Oracle, learning a control policy solely based on the proprioceptive state becomes difficult with increasing perturbation strength (Figure 3 and Appendix A.4). In all our experiments with $\sigma \in \{0.3, 0.5\}$, learning a morphology

representation improved performance (Table T4). This demonstrates, as expected, that learning an environment encoding is a viable option to deal with strongly perturbed environments.

## 5.2 The morphology encoding can be regressed from experience

The morphology parameters $c_\sigma \in C_\sigma$ influence the transition dynamics $p(\mathbf{s}_{t+1} | \mathbf{s}_t, \mathbf{a}_t, c_\sigma)$ by changing the physical properties of the body. Therefore, they are implicitly represented in a sequence of transitions, from which they might be inferred. To test this possibility, we consider a recently proposed algorithm for adaptive locomotion called rapid motor adaptation (RMA) [15]. It leverages the latent representation of the body generated by a trained Oracle, which has access to the contextual information, to train a TCN network to regress the context encoding from a short history of $T$ proprioceptive states and actions $H^{(t)} = ((\mathbf{s}^{(t-T)}, \mathbf{a}^{(t-T)}), ..., (\mathbf{s}^{(t-1)}, \mathbf{a}^{(t-1)}))$ with $T = 30$ transitions (see Appendix A.2 for hyperparameters).

Table 2: IID performance of RMA and Oracle, mean episode reward $\pm$ SEM.

| Env | Ant | | Half Cheetah | | Walker | | Hopper | |
|-----|-----|-----|-----|-----|-----|-----|-----|-----|
| Algo | RMA | Oracle | RMA | Oracle | RMA | Oracle | RMA | Oracle |
| $\sigma = 0.1$ | $2138 \pm 16$ | $2148 \pm 16$ | $2197 \pm 16$ | $2203 \pm 17$ | $1750 \pm 28$ | $1780 \pm 26$ | $1859 \pm 19$ | $1859 \pm 18$ |
| $\sigma = 0.3$ | $1700 \pm 17$ | $1723 \pm 18$ | $1402 \pm 27$ | $\mathbf{1469 \pm 24}$ | $836 \pm 23$ | $\mathbf{908 \pm 23}$ | $1267 \pm 27$ | $1296 \pm 26$ |
| $\sigma = 0.5$ | $966 \pm 16$ | $974 \pm 18$ | $595 \pm 26$ | $\mathbf{668 \pm 24}$ | $579 \pm 17$ | $\mathbf{625 \pm 17}$ | $730 \pm 21$ | $\mathbf{919 \pm 18}$ |

RMA achieves similar average reward to the Oracle (Table 2). Thus, a sequence of 30 transitions of previous proprioceptive states and actions (approx. 0.5 s in our experiments) is sufficient to extract an embedding of the context that yields high reward. This demonstrates that RMA's two-phase adaptation procedure is also viable for morphological perturbations.

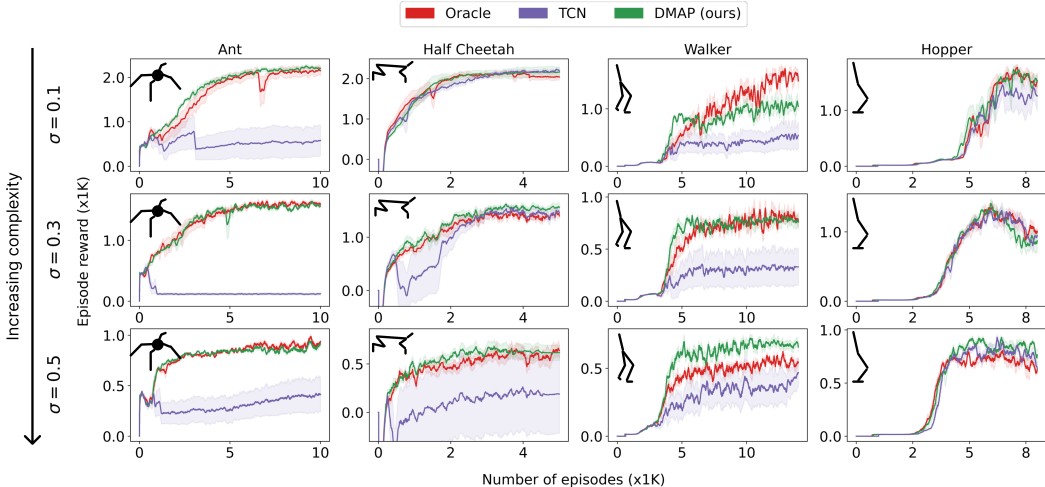

Figure 4: Learning curves of Oracle, DMAP and TCN, the network architecture used in RMA, but trained end-to-end (mean ± SEM across 5 random seeds). The performance gap between TCN and DMAP is particularly clear for Ant and Walker, in which the training of TCN is characterized by high variance and overall low reward. Overall, DMAP, without access to the perturbation state, matches the learning performance of Oracle in most environments, with the exception of Walker $\sigma = 0.1$, where Oracle is better, and Walker $\sigma = 0.5$, where DMAP is better.

## 5.3 DMAP enables end-to-end training of a morphology encoding network

Our RMA experiments demonstrated that a temporal convolutional network can regress the morphology encoding from a sequence of proprioceptive states and actions representing 0.5 seconds of transition history. However, it is unclear whether RMA's TCN encoder requires RMA's two-step training through imitation to successfully extract a morphology encoding from a sequence of transi-

tions. In fact, if such an encoding helps maximizing the reward, the TCN might successfully learn to extract the same representation by end-to-end reinforcement learning.

Remarkably, end-to-end training of a TCN encoder proves successful for the Hopper and the Half Cheetah. However, we encountered strong instability for the Ant and Walker across the random seeds (Figure 4). Surprisingly, in these two latter environments the addition of the TCN encoder is even harmful for the overall performance, as Simple achieves stronger results in the same environments (Appendix A.4). Further analysis reveals that for certain random seeds, the training process is disrupted before the agent can learn how to walk (Appendix A.4).

We hypothesize that this inability to learn is due to the difficulty of extracting a meaningful encoding to directly optimize the reward without a suitable *inductive bias*. For complex morphologies, the TCN encoder might even act as a confounder for the policy network. Indeed, replacing the TCN with DMAP leads to better stability during the training (Figure 4) and to a general performance improvement, particularly for the Ant and Walker (Appendix A.4). Differently from TCN, DMAP performs on par with Oracle in all the considered environments, with the exception of Walker with $\sigma = 0.1$. Oracle and DMAP also perform similarly when tested OOD, with a slight advantage for DMAP (Figure 5 and Appendix A.4, A.5). Finally, we compared the adaptation speed of RMA and DMAP to unseen morphological perturbations. We found that the two algorithms perform motor adaptation equally fast (Appendix A.6). Thus, DMAP learns to robustly locomote in an end-to-end fashion, without access to the morphology parameters.

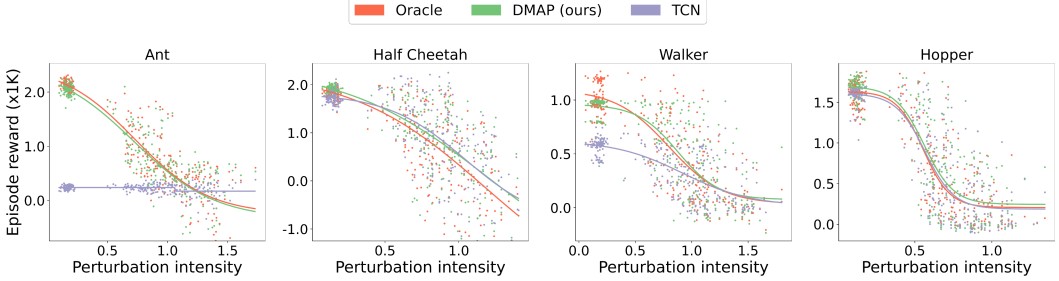

Figure 5: Reward obtained by Oracle, TCN and DMAP, trained in the $\sigma = 0.3$ environments, on a wide range of perturbations (both IID and OOD). Each point represents the average reward across 5 random seeds for one body configuration, while the perturbation intensity is the Euclidean norm of the perturbation vector. The curves are obtained by fitting a sigmoid function. Despite the larger perturbations falling outside of the training distribution, DMAP slightly outperforms Oracle, which receives the exact perturbation information. TCN performs worse for more challenging morphologies (Ant, Walker).

## 5.4 Ablation: The morphology encoding is key in statically unstable environments

To isolate the contribution of the morphology encoding on the performance of DMAP, we performed test-time ablation by evaluating the trained, distributed DMAP policy, while removing the morphology encoding. We found a performance drop for all the environments and perturbation levels (Table 3). This performance decay is especially severe in Hopper and Walker (approx. $50\%$), where the controller with ablated encoding fails to stabilize the agents, causing the early termination condition to be met sooner and thus yielding lower episode reward. This suggests that the attention mechanism is fundamental for the dynamic stability of these agents.

Table 3: IID performance of DMAP and non-encoding/ablated (ne) DMAP, mean episode reward $\pm$ SEM.

| Env | Ant | | Half Cheetah | | Walker | | Hopper | |
|---|---|---|---|---|---|---|---|---|
| Algo | DMAP | DMAP-ne | DMAP | DMAP-ne | DMAP | DMAP-ne | DMAP | DMAP-ne |
| $\sigma = 0.1$ | $\mathbf{2240 \pm 11}$ | $1966 \pm 18$ | $\mathbf{2261 \pm 11}$ | $1493 \pm 15$ | $\mathbf{1229 \pm 24}$ | $337 \pm 16$ | $\mathbf{1842 \pm 18}$ | $984 \pm 33$ |
| $\sigma = 0.3$ | $\mathbf{1623 \pm 19}$ | $1542 \pm 19$ | $\mathbf{1577 \pm 22}$ | $1132 \pm 29$ | $\mathbf{893 \pm 11}$ | $470 \pm 16$ | $\mathbf{1316 \pm 24}$ | $748 \pm 23$ |
| $\sigma = 0.5$ | $\mathbf{960 \pm 14}$ | $881 \pm 12$ | $\mathbf{669 \pm 23}$ | $507 \pm 29$ | $\mathbf{743 \pm 15}$ | $341 \pm 17$ | $\mathbf{953 \pm 15}$ | $482 \pm 20$ |

## 5.5 How does attention gate sensorimotor information?

We hypothesized that DMAP enables the agent to learn which components of the input are relevant for each joint's controller. Next, we verify that such pathways emerge by analyzing the values of the attentional matrix $K^{(t)}$, both in the *steady* and *dynamic* condition.

The steady attentional matrix $K$ represents the strength of each input to output (i.e., sensorimotor) connection. We computed morphology-specific patterns by averaging elements across time and episodes (Figure 6). As in the spinal cord, the controllers typically attend to the sensory input of mechanically linked parts (e.g., foot controller to foot velocity, etc.). Yet, this topography is not universal: for instance in the hopper, all controllers focus on the leg angle and the foot velocity. Furthermore, the attention patterns seem to reflect the bilateral symmetry of the Ant and Walker and a front-back asymmetry due to the directionality of the locomotion task. However, the attention maps developed by DMAP vary for different seeds, indeed even gait patterns can vary across seeds. Furthermore, many state and action parameters are correlated during simple locomotion. This makes interpreting why specific connections emerge in the attention maps challenging. Despite this, across seeds, we often observe bilateral and back-front symmetry. In the future, we want to study this with more challenging locomotion tasks that decorrelate the states (e.g. non-flat terrains).

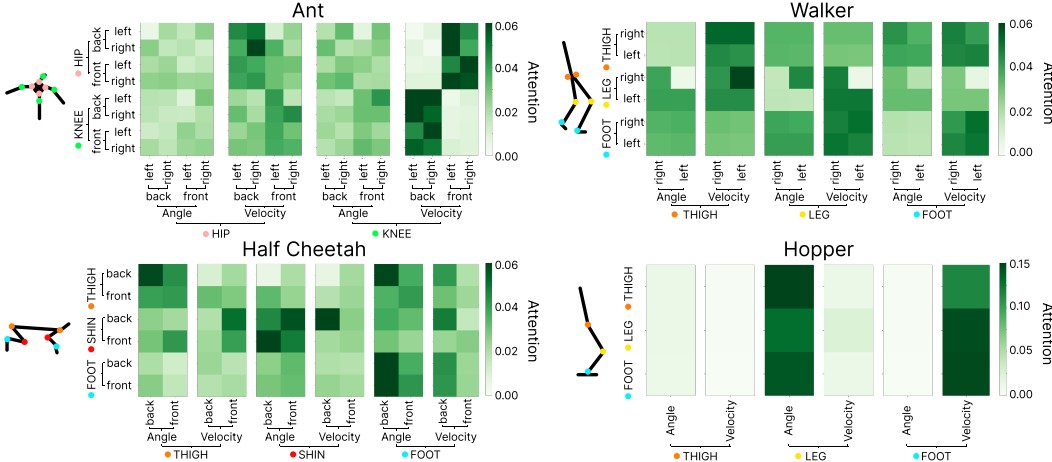

Figure 6: Attentional matrices $K$, averaged across time and IID test body perturbations, represented as a heatmap for each agent, trained with $\sigma = 0.1$. We can observe several interpretable patterns, e.g., the bilateral symmetry in the Ant, as well as a front-back asymmetry (due to the fixed running direction). Also in the Ant, the hip controllers mainly look at the angular velocity of the front knees and of the back hips, while the ankle controllers do the opposite. In the Half Cheetah, too, the attention map develops a front-back asymmetry, with a preference for the input angular position, and each controller mainly attends to the state of the joint it controls.

To analyze the temporal evolution of attention, we projected the attentional matrix $K^{(t)}$ onto a 3-dimensional space learned with UMAP [51]. Depending on the phase of the gait, the attentional matrix $K^{(t)}$ strengthens or weakens specific sensorimotor pathways (Figure 7 A). The graphs reveal a rotational dynamics of the gain modulation typical of biological, legged locomotion [52, 53, 11]. Interestingly, in DMAP the attentional dynamics seem significantly less tangled than that of the proprioceptive and action input (Figure 7 B). We quantified this visual impression across different environments and found a robust effect (Appendix A.7). We speculate that this untangling mechanism might contribute to the robustness of the policy to morphological perturbations.

## 6 Discussion

We proposed DMAP, a novel architecture that integrates computational principles of the sensorimotor system in reinforcement learning architectures. We find that DMAP enables the end-to-end training of a RL policies in environments with changing body parameters, without having access to the

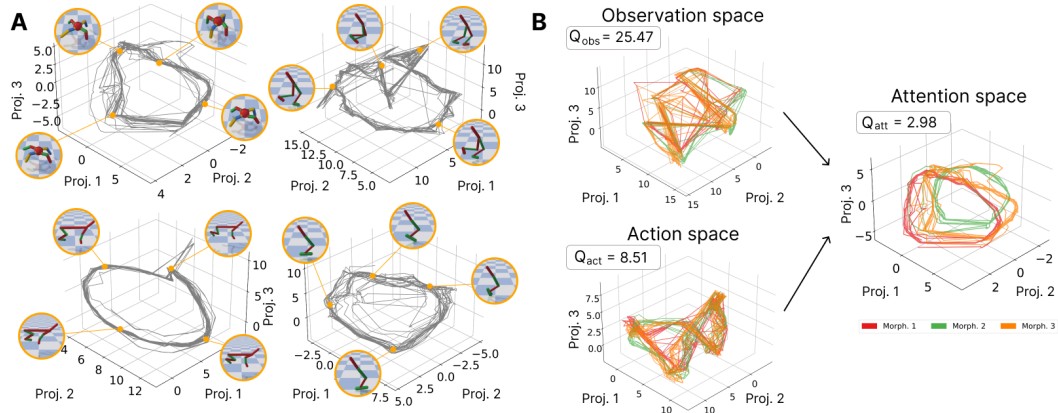

Figure 7: **A.** UMAP embedding of the attention matrix $K_t$ evolving during a single episode for each agent. The attentional dynamics of $K_t$ are cyclic, consistently with the periodic gait of the agents. Thus, the connectivity patterns between input channels and joint controllers change throughout the different phases of the gait. **B.** UMAP embeddings of the input observations $\mathcal{O}^{(t)} \in \mathbb{R}^{N_S \times T}$, input actions $\mathcal{A}^{(t)} \in \mathbb{R}^{N_A \times T}$ and attentional matrix $K_t$ for 3 different morphologies of the Ant. The change of body configuration causes a shift of the attentional dynamics in the low-dimensional space. The trajectories of the attention embedding seem less tangled than the corresponding input. We confirmed this impression by quantifying trajectories with a tangling metric $Q$ (averaged over time and morphologies). Larger values correspond to more tangled trajectories and the resulting $Q$ is shown in the plots. We refer to Appendix A.7 for more technical details.

perturbation information. Thereby, DMAP matches or even surpasses the performance of an Oracle when tested in a zero-shot manner on unseen body configurations.

Interestingly, unlike DMAP, a TCN cannot be trained end-to-end to achieve robust locomotion for more complex morphologies (Section 5.3). Our work provides an example for biological principles providing excellent inductive biases for learning, consistent with the idea that innate, structured brain connectivity nurtures biological learning [54]. Furthermore, the attention mechanism of DMAP solves this challenging task by establishing gait-phase-dependent gain modulation, robust across morphologies. Notably, synaptic gains in the nervous system also change in a phase-dependent way [52, 53, 11].

**Limitations and broader impact.** One characteristic of our current architecture design is that each joint controller observes the full proprioceptive state. A future direction will be to limit the input to the local sensory data. Such design, separating low-latency sensorimotor connection (reflexes) and slower connections to distal body parts, would be closer to the locomotor system of the animals [2]. Furthermore, in our experiments, we have used an oracle critic to learn an estimate of the cumulative reward and guide the training of the policy. This gave slightly better results, however, a suitable inductive bias (like DMAP) or possibly inspired by the neural circuits responsible for reward prediction error, could remove this (training) limitation. Indeed using purely local connectivity, the critic does not need access to the body parameters; we will describe this in a future study.

Our research aims to facilitate the training of control policies by transferring insights from neuroscience to artificial intelligence. Powerful reinforcement learning algorithms might find application in robotics, potentially enabling, in the long term, a large-scale deployment of autonomous agents. Automation comes with great opportunities, and ethical concerns [55].

## Acknowledgments and Disclosure of Funding

This research was supported by EPFL and the Swiss Government Excellence Scholarships to AMV. We thank Çağlar Gülçehre, Mackenzie Mathis, Ann Huang, Viva Berlenghi, Lucas Stoffl, Bartlomiej Borzyszkowski, Mu Zhou and Haozhe Qi for their feedback on earlier versions of this manuscript. We also thank Viva Berlenghi for contributing to the single leg perturbation experiments.

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
