# A Appendix

## A.1 DMAP detailed architecture

The Distributed Morphological Attention Policy (DMAP) consists of three main components: independent low-level proprioceptive processing, an attention-based morphology encoder and a distributed joint controller (Figure F1).

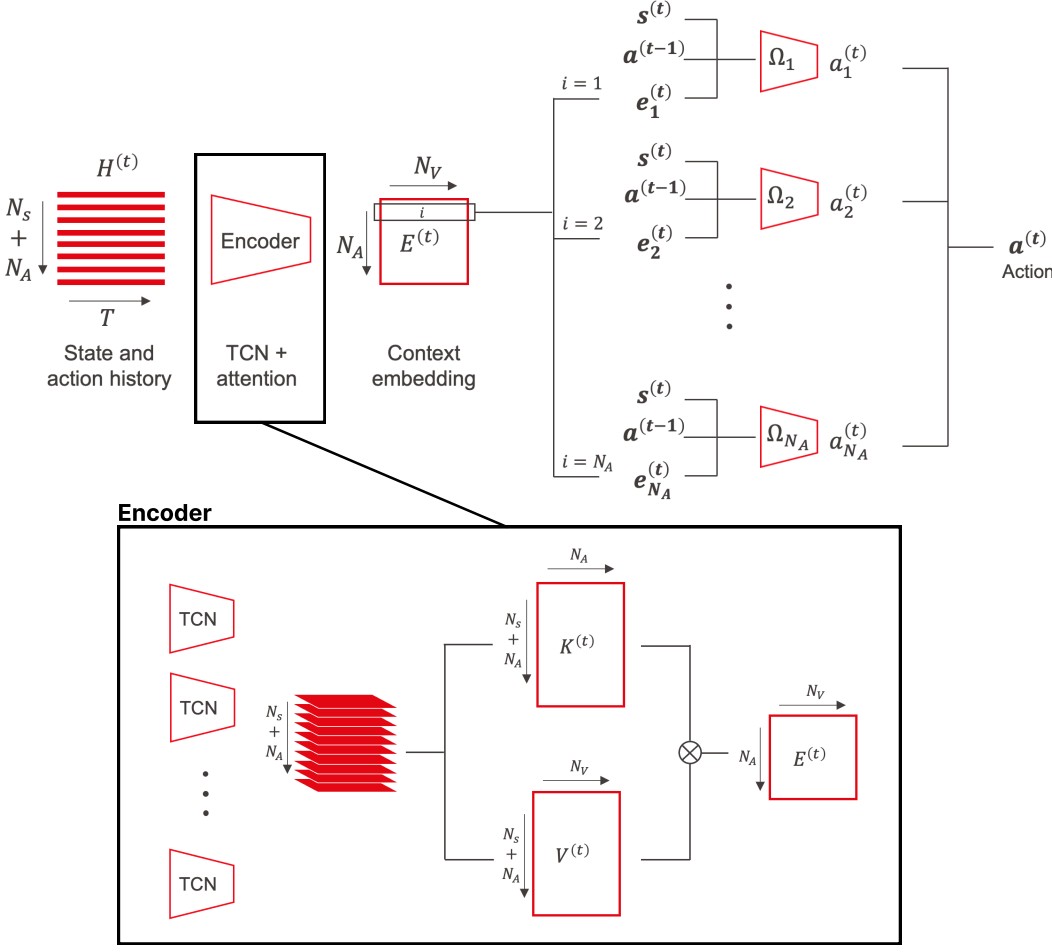

Figure F1: Detailed architecture of DMAP. After applying convolutions to each input channel along the temporal dimension, the encoder module extracts: a key matrix $K^{(t)}$, whose elements represent the attention score of an action component towards an input channel; a value matrix $V^{(t)}$, whose rows encode the information from each input channel of the transition history. A softmax operation is applied along the columns of the $K^{(t)}$ matrix, so that the total attention of one action component sums to 1. The action component-specific encoding vectors $\mathbf{e}^{(t)}$ are obtained by matrix multiplication: $K^{(t)T}V^{(t)}$. The learned context embedding is then concatenated with the previous action $\mathbf{a}^{(t-1)}$ and the current proprioceptive observation $\mathbf{s}^{(t)}$. Together, those triples are the inputs to $N_A$ fully-connected networks, one for each action component. They output the torque applied to one joint of the agent.

## A.2 Hyperparameters

Table T1 lists the hyperparameters of SAC used for our experiments. They are the same for all network architectures, without specific fine tuning. We also list the parameters of the neural networks of the different agents.

Table T1: Parameters of SAC for each of the 4 environments

| Environment Algorithms | Parameters | Ant | Half Cheetah | Walker | Hopper |
|---|---|---|---|---|---|
| All | Action normalization | Yes | Yes | Yes | Yes |
| | Num TD steps | 1 | 1 | 1 | 1 |
| | Buffer size | 1 000 000 | 300 000 | 300 000 | 300 000 |
| | Prioritized experience replay | No | Yes | Yes | Yes |
| | Actor learning rate | 0.0003 | 0.0003 | 0.0003 | 0.0003 |
| | Critic learning rate | 0.0003 | 0.0003 | 0.0003 | 0.0003 |
| | Entropy learning rate | 0.0003 | 0.0003 | 0.0003 | 0.0003 |
| | Discount factor $\gamma$ | 0.995 | 0.995 | 0.995 | 0.995 |
| | Minibatch size | 256 | 256 | 256 | 256 |
| Simple | Policy hiddens | [256, 256] | [256, 256] | [256, 256] | [256, 256] |
| | Q hiddens | [256, 256] | [256, 256] | [256, 256] | [256, 256] |
| | Activation | ReLU | ReLU | ReLU | ReLU |
| Oracle | Policy encoder hiddens | [256, 128] | [256, 128] | [256, 128] | [256, 128] |
| | Encoding size | 4 | 4 | 4 | 4 |
| | Policy hiddens | [128, 128] | [128, 128] | [128, 128] | [128, 128] |
| | Q encoder hiddens | [256, 128] | [256, 128] | [256, 128] | [256, 128] |
| | Q hiddens | [128, 128] | [128, 128] | [128, 128] | [128, 128] |
| | Activation | ReLU | ReLU | ReLU | ReLU |
| RMA/TCN | Num TCN layers | 3 | 3 | 3 | 3 |
| | Num filters per layer | [32, 32, 32] | [32, 32, 32] | [32, 32, 32] | [32, 32, 32] |
| | Kernel size per layer | [5, 3, 3] | [5, 3, 3] | [5, 3, 3] | [5, 3, 3] |
| | Stride per layer | [4, 1, 1] | [4, 1, 1] | [4, 1, 1] | [4, 1, 1] |
| | Policy encoder hiddens | [128, 32] | [128, 32] | [128, 32] | [128, 32] |
| | Encoding size | 4 | 4 | 4 | 4 |
| | Policy hiddens | [128, 128] | [128, 128] | [128, 128] | [128, 128] |
| | Q encoder hiddens | [256, 128] | [256, 128] | [256, 128] | [256, 128] |
| | Q hiddens | [128, 128] | [128, 128] | [128, 128] | [128, 128] |
| | Activation | ReLU | ReLU | ReLU | ReLU |
| | History size T | 30 | 30 | 30 | 30 |
| DMAP | Num TCN layers | 3 | 3 | 3 | 3 |
| | Num filters per layer | [32, 32, 32] | [32, 32, 32] | [32, 32, 32] | [32, 32, 32] |
| | Kernel size per layer | [5, 3, 3] | [5, 3, 3] | [5, 3, 3] | [5, 3, 3] |
| | Stride per layer | [4, 1, 1] | [4, 1, 1] | [4, 1, 1] | [4, 1, 1] |
| | Policy encoder hiddens | [128, 32] | [128, 32] | [128, 32] | [128, 32] |
| | Encoding size | 4 | 4 | 4 | 4 |
| | Policy hiddens | [32, 32] | [32, 32] | [32, 32] | [32, 32] |
| | Num policy networks | 8 | 6 | 6 | 3 |
| | Q encoder hiddens | [256, 128] | [256, 128] | [256, 128] | [256, 128] |
| | Q hiddens | [128, 128] | [128, 128] | [128, 128] | [128, 128] |
| | Activation | ReLU | ReLU | ReLU | ReLU |
| | History size T | 30 | 30 | 30 | 30 |

## A.3 Perturbed environments specifications

The state and action components of the four perturbed environments (Ant, Half Cheetah, Walker, Hopper) are the same as the ones of the environment they are derived from. They are listed in Table T2, ordered as they appear in the state and action vectors. The body parameters are sampled from a uniform distribution at the beginning of each episode. The body segments are perturbed in small subgroups, e.g., the two segments forming a limb of Ant, in which the perturbation is consistent. The parameters defining the perturbation space are detailed in Table T3.

During our experiments, we experienced an unusual behavior in the Half Cheetah environment for particularly light limbs (i.e., ones with small thickness). In this case, the Half Cheetah could move the legs at very high speed, allowing an unnatural locomotion in PyBullet, in which the agent could advance without touching the ground, seemingly ignoring gravity. For this reason, we set a lower bound to the perturbation on the length and the thickness of the limbs, corresponding to $-0.3$ (only for the Half Cheetah with $\sigma = 0.5$). This clipping removed the unusual behavior of the simulator.

Table T2: State and action components of each environment. $\alpha$ is the angle between the target direction and the current running direction. $v_i$, $i \in x, y, z$ are the components of the linear velocity vector.

| Environment Components | Ant | Half Cheetah | Walker | Hopper |
|---|---|---|---|---|
| State | Elevation | Elevation | Elevation | Elevation |
| | $sin(\alpha)$ | $sin(\alpha)$ | $sin(\alpha)$ | $sin(\alpha)$ |
| | $cos(\alpha)$ | $cos(\alpha)$ | $cos(\alpha)$ | $cos(\alpha)$ |
| | $v_x$ | $v_x$ | $v_x$ | $v_x$ |
| | $v_y$ | $v_y$ | $v_y$ | $v_y$ |
| | $v_z$ | $v_z$ | $v_z$ | $v_z$ |
| | roll | roll | roll | roll |
| | pitch | pitch | pitch | pitch |
| | Back left hip angle | Back thigh angle | Right thigh angle | Thigh angle |
| | Back left hip velocity | Back thigh velocity | Right thigh velocity | Thigh velocity |
| | Back left knee angle | Back shin angle | Right leg angle | Leg angle |
| | Back left knee velocity | Back shin velocity | Right leg velocity | Leg velocity |
| | Front left hip angle | Back foot angle | Right foot angle | Foot angle |
| | Front left hip velocity | Back foot velocity | Right foot velocity | Foot velocity |
| | Front left knee angle | Front thigh angle | Left thigh angle | Foot contact |
| | Front left knee velocity | Front thigh velocity | Left thigh velocity | |
| | Back right hip angle | Front foot angle | Left leg angle | |
| | Back right hip velocity | Front foot velocity | Left leg velocity | |
| | Back right knee angle | Front foot contact | Left foot angle | |
| | Back right knee velocity | Front shin contact | Left foot velocity | |
| | Front right hip angle | Front thigh contact | Right foot contact | |
| | Front right hip velocity | Back foot contact | Left foot contact | |
| | Front right knee angle | Back shin contact | | |
| | Front right knee velocity | Back thigh contact | | |
| | Back left foot contact | | | |
| | Front left foot contact | | | |
| | Back right foot contact | | | |
| | Front right foot contact | | | |
| Action | Back left hip torque | Front foot torque | Right thigh torque | Thigh torque |
| | Back left knee torque | Front shin torque | Right leg torque | Leg torque |
| | Front left hip torque | Front thigh torque | Right foot torque | Foot torque |
| | Front left knee torque | Back foot torque | Left thigh torque | |
| | Back right hip torque | Back shin torque | Left leg torque | |
| | Back right knee torque | Back thigh torque | Left foot torque | |
| | Front right hip torque | | | |
| | Front right knee torque | | | |

## A.4 Performance summary

Tables T4 and T5 compare the IID performance of Simple-Oracle and of TCN-DMAP. The numbers in bold denote a significant statistical difference between the two methods (p-value $< 0.001$, paired t-test).

We also list the IID (Table T6) and OOD (Tables T7, T8 and T9) test results of all the agents trained for this work. Some negative values should not surprise the reader, as some agents, when tested way outside of the training distribution, fail to walk, collecting more penalties (e.g., due to undesired contact force or excessive energy expenditure) than positive reward.

We also show the graphs of the reward as a function for different perturbation intensity for the end-to-end trained Oracle, DMAP and TCN (Figure F2). Generally, DMAP performs similarly to the Oracle, while the TCN has lower performance especially for more challenging morphologies (Ant, Walker).

Table T3: Perturbation parameters of each environment.

| Environment
Perturbation type | Ant | Half Cheetah | Walker | Hopper |
|---|---|---|---|---|
| Thickness | Torso
Front left limb
Front right limb
Back left limb
Back right limb | Head
Torso
Back limb
Front limb | Torso
Limbs
Feet | Torso
Limb
Foot |
| Length | Front left limb
Front right limb
Back left limb
Back right limb | Torso
Back limb
Front limb | Limbs
Feet | Limb
Foot |
| Total | 9 | 7 | 5 | 5 |

Table T4: IID performance of Simple and Oracle, mean episode reward $\pm$ SEM across 5 different random seeds, on 100 body configurations not encountered during the training.

| Env | Ant | | Half Cheetah | | Walker | | Hopper | |
|---|---|---|---|---|---|---|---|---|
| Algo | Simple | Oracle | Simple | Oracle | Simple | Oracle | Simple | Oracle |
| $\sigma = 0.1$ | $2164 \pm 17$ | $2148 \pm 16$ | $1633 \pm 25$ | $\mathbf{2203 \pm 17}$ | $\mathbf{1909 \pm 17}$ | $1780 \pm 26$ | $1807 \pm 25$ | $1859 \pm 18$ |
| $\sigma = 0.3$ | $1270 \pm 20$ | $\mathbf{1723 \pm 18}$ | $1412 \pm 22$ | $\mathbf{1469 \pm 24}$ | $691 \pm 17$ | $\mathbf{908 \pm 23}$ | $1121 \pm 18$ | $\mathbf{1296 \pm 26}$ |
| $\sigma = 0.5$ | $391 \pm 14$ | $\mathbf{974 \pm 18}$ | $482 \pm 19$ | $\mathbf{668 \pm 24}$ | $397 \pm 14$ | $\mathbf{625 \pm 17}$ | $863 \pm 19$ | $\mathbf{919 \pm 18}$ |

Table T5: IID performance of TCN and DMAP, mean episode reward $\pm$ SEM.

| Env | Ant | | Half Cheetah | | Walker | | Hopper | |
|---|---|---|---|---|---|---|---|---|
| Algo | TCN | DMAP | TCN | DMAP | TCN | DMAP | TCN | DMAP |
| $\sigma = 0.1$ | $932 \pm 45$ | $\mathbf{2240 \pm 11}$ | $2278 \pm 10$ | $2261 \pm 11$ | $1060 \pm 42$ | $\mathbf{1229 \pm 24}$ | $1762 \pm 16$ | $\mathbf{1842 \pm 18}$ |
| $\sigma = 0.3$ | $251 \pm 11$ | $\mathbf{1623 \pm 19}$ | $1540 \pm 20$ | $1577 \pm 22$ | $518 \pm 19$ | $\mathbf{893 \pm 11}$ | $\mathbf{1368 \pm 20}$ | $1316 \pm 24$ |
| $\sigma = 0.5$ | $481 \pm 20$ | $\mathbf{960 \pm 14}$ | $553 \pm 27$ | $\mathbf{669 \pm 23}$ | $584 \pm 18$ | $\mathbf{743 \pm 15}$ | $1017 \pm 18$ | $953 \pm 15$ |

Table T6: IID performance (train $\sigma$ = test $\sigma$) for all architectures, environments and perturbation intensities, as mean $\pm$ sem.

| Algorithm
$\sigma$ | RMA | DMAP | DMAP-ne | TCN | Oracle | Simple |
|---|---|---|---|---|---|---|
| Ant $\sigma = 0.1$ | $2138 \pm 16$ | $2240 \pm 11$ | $1966 \pm 18$ | $932 \pm 45$ | $2148 \pm 16$ | $2164 \pm 17$ |
| Ant $\sigma = 0.3$ | $1700 \pm 17$ | $1623 \pm 19$ | $1542 \pm 19$ | $251 \pm 11$ | $1723 \pm 18$ | $1270 \pm 20$ |
| Ant $\sigma = 0.5$ | $966 \pm 16$ | $960 \pm 14$ | $881 \pm 12$ | $481 \pm 20$ | $974 \pm 18$ | $391 \pm 14$ |
| Half Cheetah $\sigma = 0.1$ | $2197 \pm 16$ | $2261 \pm 11$ | $1493 \pm 15$ | $2278 \pm 10$ | $2203 \pm 17$ | $1633 \pm 25$ |
| Half Cheetah $\sigma = 0.3$ | $1402 \pm 27$ | $1577 \pm 22$ | $1132 \pm 29$ | $1540 \pm 20$ | $1469 \pm 24$ | $1412 \pm 22$ |
| Half Cheetah $\sigma = 0.5$ | $595 \pm 26$ | $669 \pm 23$ | $507 \pm 29$ | $553 \pm 27$ | $668 \pm 24$ | $482 \pm 19$ |
| Walker $\sigma = 0.1$ | $1750 \pm 28$ | $1229 \pm 24$ | $337 \pm 16$ | $1060 \pm 42$ | $1780 \pm 26$ | $1909 \pm 17$ |
| Walker $\sigma = 0.3$ | $836 \pm 23$ | $893 \pm 11$ | $470 \pm 16$ | $518 \pm 19$ | $908 \pm 23$ | $691 \pm 17$ |
| Walker $\sigma = 0.5$ | $579 \pm 17$ | $743 \pm 15$ | $341 \pm 17$ | $584 \pm 18$ | $625 \pm 17$ | $397 \pm 14$ |
| Hopper $\sigma = 0.1$ | $1859 \pm 19$ | $1842 \pm 18$ | $984 \pm 33$ | $1762 \pm 16$ | $1859 \pm 18$ | $1807 \pm 25$ |
| Hopper $\sigma = 0.3$ | $1267 \pm 27$ | $1316 \pm 24$ | $748 \pm 23$ | $1368 \pm 20$ | $1296 \pm 26$ | $1121 \pm 18$ |
| Hopper $\sigma = 0.5$ | $730 \pm 21$ | $953 \pm 15$ | $482 \pm 20$ | $1017 \pm 18$ | $919 \pm 18$ | $863 \pm 19$ |

Furthermore, we visualize the distribution of the IID and OOD episode rewards for different test morphologies for DMAP and TCN. To compare morphologies that might reach different rewards, we normalized the episode reward by the Oracle reward for each morphology (Figures F3 and F4). This highlights training challenges for the TCN on specific IID test morphologies (Figures F3), and that DMAP generalizes better than TCN to OOD test morphologies (Figure F4).

We performed one additional test-time ablation experiment, in which we removed the controller-specific weighting of the morphology encoding, which we named DMAP-nw (non-weighted). Differently from DMAP-ne, in these evaluations the agent is provided access to the morphology encoding,

Table T7: OOD performance for all architectures (train sigma 0.1), as mean $\pm$ sem

| Algorithm $\sigma$ | RMA | DMAP | DMAP-ne | TCN | Oracle | Simple |
|---|---|---|---|---|---|---|
| Ant $\sigma = 0.1$ | $2138 \pm 16$ | $2240 \pm 11$ | $1966 \pm 18$ | $932 \pm 45$ | $2148 \pm 16$ | $2164 \pm 17$ |
| Ant $\sigma = 0.3$ | $1124 \pm 26$ | $1105 \pm 26$ | $1004 \pm 22$ | $479 \pm 26$ | $1078 \pm 27$ | $1034 \pm 28$ |
| Ant $\sigma = 0.5$ | $604 \pm 17$ | $579 \pm 18$ | $536 \pm 16$ | $289 \pm 15$ | $598 \pm 16$ | $572 \pm 19$ |
| Ant $\sigma = 0.7$ | $145 \pm 30$ | $129 \pm 30$ | $139 \pm 28$ | $162 \pm 16$ | $154 \pm 30$ | $169 \pm 26$ |
| Half Cheetah $\sigma = 0.1$ | $2197 \pm 16$ | $2261 \pm 11$ | $1493 \pm 15$ | $2278 \pm 10$ | $2203 \pm 17$ | $1633 \pm 25$ |
| Half Cheetah $\sigma = 0.3$ | $1159 \pm 38$ | $1369 \pm 38$ | $877 \pm 36$ | $1426 \pm 36$ | $966 \pm 43$ | $825 \pm 36$ |
| Half Cheetah $\sigma = 0.5$ | $216 \pm 49$ | $374 \pm 52$ | $137 \pm 46$ | $475 \pm 51$ | $100 \pm 48$ | $208 \pm 41$ |
| Half Cheetah $\sigma = 0.7$ | $-307 \pm 45$ | $-142 \pm 48$ | $-205 \pm 44$ | $-147 \pm 48$ | $-375 \pm 44$ | $-225 \pm 42$ |
| Walker $\sigma = 0.1$ | $1750 \pm 28$ | $1229 \pm 24$ | $337 \pm 16$ | $1060 \pm 42$ | $1780 \pm 26$ | $1909 \pm 17$ |
| Walker $\sigma = 0.3$ | $877 \pm 37$ | $666 \pm 25$ | $283 \pm 15$ | $504 \pm 31$ | $622 \pm 34$ | $900 \pm 37$ |
| Walker $\sigma = 0.5$ | $318 \pm 26$ | $265 \pm 19$ | $153 \pm 13$ | $211 \pm 20$ | $226 \pm 20$ | $354 \pm 27$ |
| Walker $\sigma = 0.7$ | $165 \pm 20$ | $140 \pm 15$ | $114 \pm 12$ | $113 \pm 14$ | $106 \pm 14$ | $165 \pm 19$ |
| Hopper $\sigma = 0.1$ | $1859 \pm 19$ | $1842 \pm 18$ | $984 \pm 33$ | $1762 \pm 16$ | $1859 \pm 18$ | $1807 \pm 25$ |
| Hopper $\sigma = 0.3$ | $1044 \pm 34$ | $857 \pm 33$ | $536 \pm 27$ | $960 \pm 36$ | $947 \pm 33$ | $760 \pm 35$ |
| Hopper $\sigma = 0.5$ | $356 \pm 24$ | $308 \pm 23$ | $216 \pm 20$ | $310 \pm 29$ | $417 \pm 25$ | $303 \pm 22$ |
| Hopper $\sigma = 0.7$ | $217 \pm 19$ | $208 \pm 19$ | $160 \pm 15$ | $206 \pm 24$ | $219 \pm 18$ | $188 \pm 18$ |

Table T8: OOD performance for all architectures (train sigma 0.3), as mean $\pm$ sem

| Algorithm $\sigma$ | RMA | DMAP | DMAP-ne | TCN | Oracle | Simple |
|---|---|---|---|---|---|---|
| Ant $\sigma = 0.1$ | $2103 \pm 6$ | $2051 \pm 7$ | $2015 \pm 6$ | $238 \pm 12$ | $2124 \pm 5$ | $1671 \pm 20$ |
| Ant $\sigma = 0.3$ | $1700 \pm 17$ | $1623 \pm 19$ | $1542 \pm 19$ | $251 \pm 11$ | $1723 \pm 18$ | $1270 \pm 20$ |
| Ant $\sigma = 0.5$ | $924 \pm 22$ | $856 \pm 22$ | $833 \pm 20$ | $227 \pm 9$ | $900 \pm 24$ | $796 \pm 18$ |
| Ant $\sigma = 0.7$ | $306 \pm 32$ | $228 \pm 33$ | $230 \pm 32$ | $185 \pm 12$ | $261 \pm 30$ | $283 \pm 28$ |
| Half Cheetah $\sigma = 0.1$ | $1795 \pm 11$ | $1900 \pm 9$ | $1578 \pm 15$ | $1719 \pm 8$ | $1824 \pm 9$ | $1704 \pm 11$ |
| Half Cheetah $\sigma = 0.3$ | $1402 \pm 27$ | $1577 \pm 22$ | $1132 \pm 29$ | $1540 \pm 20$ | $1469 \pm 24$ | $1412 \pm 22$ |
| Half Cheetah $\sigma = 0.5$ | $822 \pm 42$ | $1033 \pm 38$ | $552 \pm 42$ | $1117 \pm 35$ | $892 \pm 43$ | $787 \pm 40$ |
| Half Cheetah $\sigma = 0.7$ | $350 \pm 48$ | $453 \pm 47$ | $151 \pm 45$ | $462 \pm 45$ | $250 \pm 50$ | $267 \pm 45$ |
| Walker $\sigma = 0.1$ | $906 \pm 22$ | $943 \pm 9$ | $482 \pm 15$ | $574 \pm 19$ | $1028 \pm 21$ | $846 \pm 12$ |
| Walker $\sigma = 0.3$ | $836 \pm 23$ | $893 \pm 11$ | $470 \pm 16$ | $518 \pm 19$ | $908 \pm 23$ | $691 \pm 17$ |
| Walker $\sigma = 0.5$ | $487 \pm 23$ | $518 \pm 20$ | $239 \pm 14$ | $342 \pm 18$ | $491 \pm 24$ | $388 \pm 18$ |
| Walker $\sigma = 0.7$ | $248 \pm 18$ | $289 \pm 19$ | $171 \pm 14$ | $219 \pm 16$ | $267 \pm 19$ | $221 \pm 16$ |
| Hopper $\sigma = 0.1$ | $1683 \pm 16$ | $1676 \pm 14$ | $729 \pm 25$ | $1591 \pm 12$ | $1618 \pm 16$ | $1351 \pm 9$ |
| Hopper $\sigma = 0.3$ | $1267 \pm 27$ | $1316 \pm 24$ | $748 \pm 23$ | $1368 \pm 20$ | $1296 \pm 26$ | $1121 \pm 18$ |
| Hopper $\sigma = 0.5$ | $532 \pm 28$ | $729 \pm 30$ | $456 \pm 23$ | $672 \pm 29$ | $666 \pm 29$ | $562 \pm 23$ |
| Hopper $\sigma = 0.7$ | $329 \pm 24$ | $443 \pm 27$ | $334 \pm 21$ | $382 \pm 26$ | $371 \pm 26$ | $342 \pm 21$ |

but all the encoding vectors of the matrix $V$ are assigned equal importance. We found that DMAP performs significantly better than DMAP-nw (which is much better than DMAP-ne), corroborating the importance of per-controller attention weights (Table T10).

Finally, based on the feedback of one reviewer, we considered a state of the art meta-learning algorithm, PEARL [1]. We test its performance compared to RMA and DMAP. As PEARL is a meta-learning algorithm, not optimized for zero-shot adaptation, we modified the training and test setup by including a fixed number of training morphologies (100), which serve as the meta-training environment. The algorithm is then tested on the same 100 test morphologies used to evaluate the other algorithms presented in the paper, but only after 10 adaptation episodes. However, we found that after 7 days of training (by comparison, DMAP requires 1 day to complete the training on the same hardware) PEARL had only achieved 664 meta-test reward in the Ant environment with $\sigma = 0.1$ and 436 meta-test reward in the Ant environment with $\sigma = 0.5$. These preliminary results are lower to those obtained with RMA and DMAP, suggesting that PEARL is not an effective algorithm to train a policy to control agents with variable morphologies. However, future work should study this in more detail (incl. optimizing hyperparameters) and in particular test if the number of training morphologies was too small for PEARL.

Table T9: OOD performance for all architectures (train sigma 0.5), as mean $\pm$ sem

| Algorithm $\sigma$ | RMA | DMAP | DMAP-ne | TCN | Oracle | Simple |
|---|---|---|---|---|---|---|
| Ant $\sigma = 0.1$ | $1667 \pm 5$ | $1477 \pm 5$ | $1347 \pm 7$ | $692 \pm 30$ | $1665 \pm 6$ | $358 \pm 13$ |
| Ant $\sigma = 0.3$ | $1428 \pm 11$ | $1280 \pm 11$ | $1133 \pm 11$ | $620 \pm 26$ | $1454 \pm 11$ | $444 \pm 14$ |
| Ant $\sigma = 0.5$ | $966 \pm 16$ | $960 \pm 14$ | $881 \pm 12$ | $481 \pm 20$ | $974 \pm 18$ | $391 \pm 14$ |
| Ant $\sigma = 0.7$ | $409 \pm 27$ | $504 \pm 22$ | $364 \pm 31$ | $273 \pm 21$ | $479 \pm 21$ | $236 \pm 20$ |
| Half Cheetah $\sigma = 0.1$ | $748 \pm 23$ | $816 \pm 12$ | $718 \pm 18$ | $466 \pm 33$ | $834 \pm 17$ | $601 \pm 7$ |
| Half Cheetah $\sigma = 0.3$ | $697 \pm 23$ | $788 \pm 15$ | $660 \pm 22$ | $603 \pm 27$ | $766 \pm 21$ | $585 \pm 9$ |
| Half Cheetah $\sigma = 0.5$ | $595 \pm 26$ | $669 \pm 23$ | $507 \pm 29$ | $553 \pm 27$ | $668 \pm 24$ | $482 \pm 19$ |
| Half Cheetah $\sigma = 0.7$ | $399 \pm 30$ | $443 \pm 30$ | $303 \pm 33$ | $373 \pm 31$ | $364 \pm 33$ | $312 \pm 27$ |
| Walker $\sigma = 0.1$ | $746 \pm 13$ | $949 \pm 7$ | $533 \pm 17$ | $702 \pm 16$ | $783 \pm 14$ | $468 \pm 12$ |
| Walker $\sigma = 0.3$ | $722 \pm 14$ | $887 \pm 11$ | $490 \pm 17$ | $716 \pm 16$ | $787 \pm 13$ | $484 \pm 13$ |
| Walker $\sigma = 0.5$ | $579 \pm 17$ | $743 \pm 15$ | $341 \pm 17$ | $584 \pm 18$ | $625 \pm 17$ | $397 \pm 14$ |
| Walker $\sigma = 0.7$ | $374 \pm 17$ | $504 \pm 19$ | $207 \pm 15$ | $418 \pm 19$ | $424 \pm 18$ | $318 \pm 14$ |
| Hopper $\sigma = 0.1$ | $1169 \pm 13$ | $1109 \pm 5$ | $612 \pm 19$ | $1272 \pm 8$ | $1224 \pm 13$ | $1002 \pm 7$ |
| Hopper $\sigma = 0.3$ | $961 \pm 18$ | $1102 \pm 8$ | $675 \pm 19$ | $1161 \pm 14$ | $1070 \pm 16$ | $1036 \pm 12$ |
| Hopper $\sigma = 0.5$ | $730 \pm 21$ | $953 \pm 15$ | $482 \pm 20$ | $1017 \pm 18$ | $919 \pm 18$ | $863 \pm 19$ |
| Hopper $\sigma = 0.7$ | $428 \pm 21$ | $591 \pm 23$ | $342 \pm 19$ | $643 \pm 25$ | $558 \pm 22$ | $576 \pm 22$ |

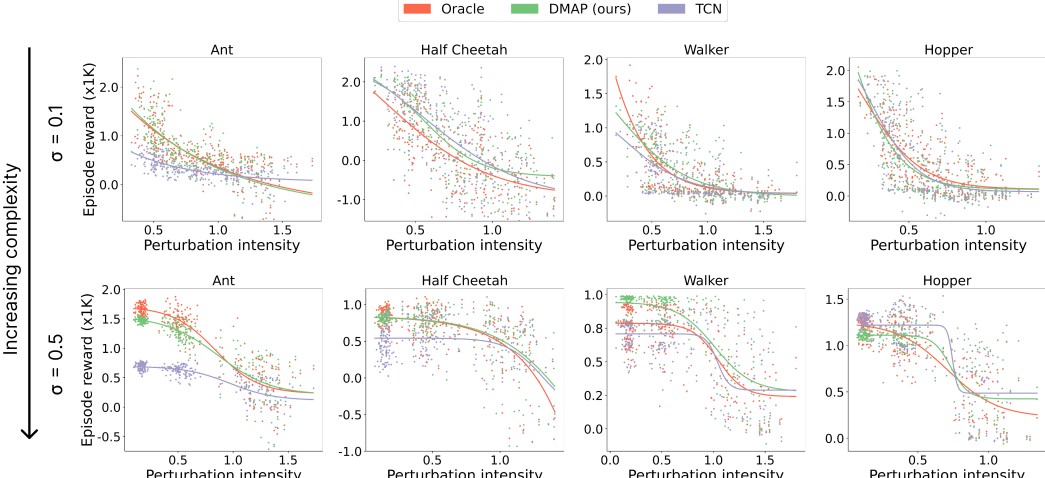

Figure F2: Reward obtained by Oracle, TCN and DMAP, trained in the $\sigma = 0.1$ and $\sigma = 0.5$ environments, on a wide range of perturbations (both IID and OOD). Each point represents the average reward across 5 random seeds for one body configuration, while the perturbation intensity is the eucledian norm of the perturbation vector. The curves are obtained by fitting a sigmoid function. DMAP matches or even surpasses the Oracle performance and outperform TCN (except for Hopper with $\sigma = 0.5$) for all training environments.

Table T10: IID performance of TCN and DMAP, mean episode reward $\pm$ SEM.

| Env Algo $\sigma$ | Ant DMAP | DMAP-nw | Half Cheetah DMAP | DMAP-nw | Walker DMAP | DMAP-nw | Hopper DMAP | DMAP-nw |
|---|---|---|---|---|---|---|---|---|
| 0.1 | $\mathbf{2240 \pm 11}$ | $2178 \pm 10$ | $\mathbf{2261 \pm 11}$ | $2185 \pm 13$ | $\mathbf{1229 \pm 24}$ | $1041 \pm 29$ | $\mathbf{1842 \pm 18}$ | $1179 \pm 28$ |
| 0.3 | $\mathbf{1623 \pm 19}$ | $1563 \pm 19$ | $\mathbf{1577 \pm 22}$ | $1439 \pm 23$ | $\mathbf{893 \pm 11}$ | $858 \pm 14$ | $\mathbf{1316 \pm 24}$ | $1014 \pm 28$ |
| 0.5 | $\mathbf{960 \pm 14}$ | $899 \pm 14$ | $\mathbf{669 \pm 23}$ | $572 \pm 21$ | $\mathbf{743 \pm 15}$ | $698 \pm 22$ | $\mathbf{953 \pm 15}$ | $777 \pm 18$ |

## A.5   Robustness to a single leg length perturbation

To quantify the robustness of DMAP to a specific morphological perturbation, we performed an experiment where we progressively change the length of a limb in the ant (from 0% perturbation

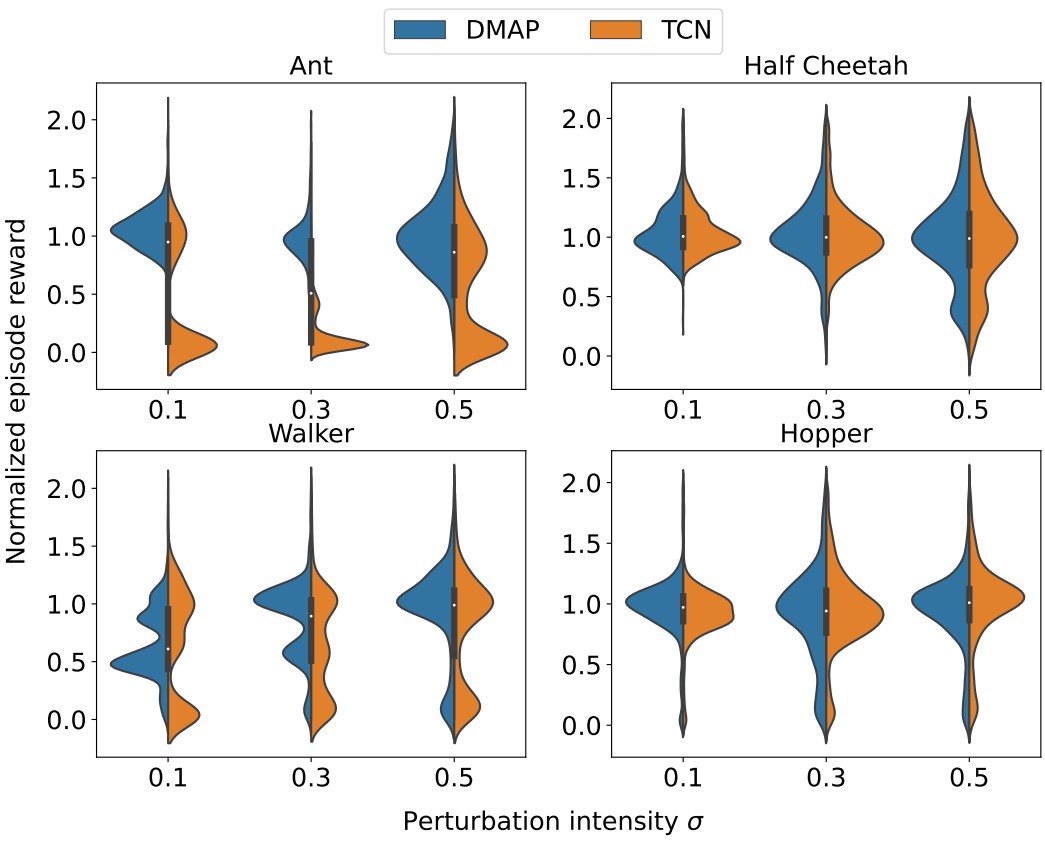

Figure F3: The normalized cumulative episode reward collected in each of the 100 test body configurations (IID test) by TCN and DMAP. Normalization was done based on Oracle rewards per morphology. We can observe that the low reward obtained by TCN in Ant and Walker is due to the density peaks in the lower part of the graphs, corresponding to agents trained with random seeds, which did not lead to a successful outcome.

intensity, which is the normal limb, to 100%, which corresponds to limb amputation). We investigated how it affects the episode reward.

We found that the reward monotonically decreases, with a similar decay independent which limb is perturbed (Figure F5). Furthermore, we found an increase in robustness when DMAP is trained with higher perturbation intensity (Figure F5).

## A.6 Adaptation speed to new morphological perturbation

One distinctive characteristic of RMA [2] is the short time it requires to perform motor adaptation to perturbations. We sought to compare the adaptation speed for RMA and DMAP. We could not perform online changes of the morphological parameters within an episode in PyBullet, therefore we compared speeds of RMA and DMAP when starting to run from the resting position with an unseen morphology in the Ant environment throughout all test episodes. We define the average speed as the norm of the projection of the velocity vector of the agent's center of mass on the running surface and, since it varies during one gait cycle, we average it over a window of 30 transitions. We find that for all perturbation intensities DMAP and RMA reach the final speed after a similar number of timesteps (Figure F6). This analysis suggests that the difference in the training procedure (imitation of the Oracle's encoder for RMA, end-to-end for DMAP) does not lead to a relevant changes in adaptation speed. Furthermore, it shows that DMAP can also rapidly adapt.

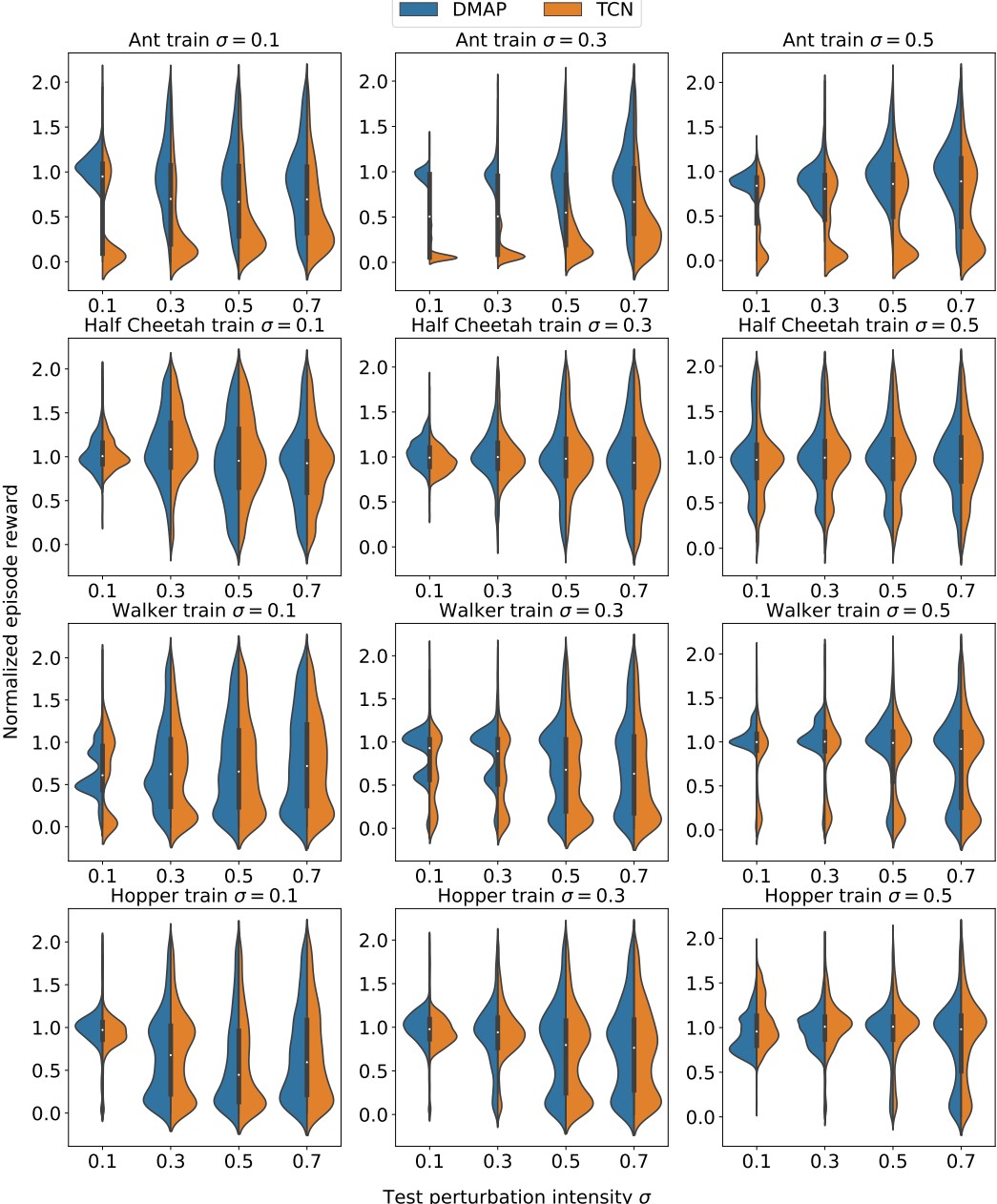

Figure F4: As in Figure F3, but keeping the training $\sigma$ fixed, while changing the test $\sigma$. In general, as the perturbation level increases, the distributions become less concentrated, because the variance of the relative performance increases. However, this suggests that OOD performance of the Oracle is a poor indicator for the generalization complexity of a morphology.

### A.7 Analysis of the attentional dynamics

When we analyzed the attentional dynamics, we found rotational dynamics and less tangling of the attentional dynamics than of the observation or action spaces for a few (randomly selected) episodes of the ant. We quantified the tangling of a trajectory with the measure introduced by Russo et al. [3]:

$$Q(t) = \max_{t'} \frac{||\dot{x}_t - \dot{x}_{t'}||^2}{||x_t - x_{t'}||^2 + \epsilon},$$

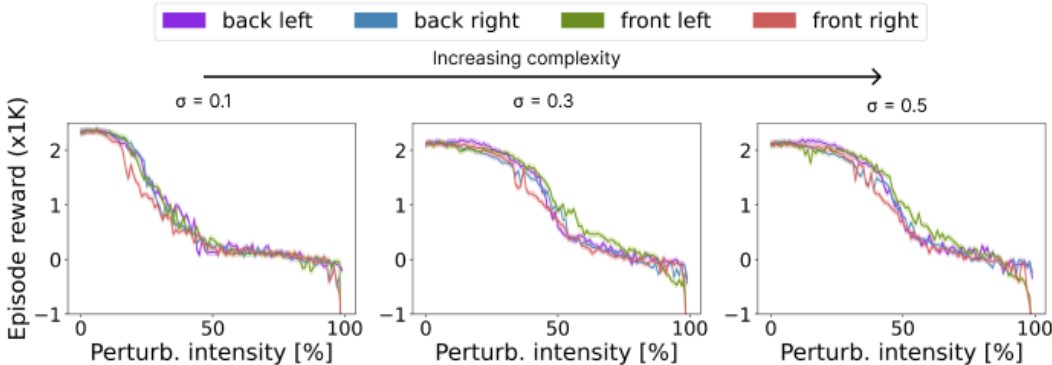

Figure F5: Reward obtained by DMAP when the length of a specific limb of the Ant agent is decreased until breakage (mean ± SEM across 5 random seeds). The perturbation intensity is expressed as the percentage of perturbation applied to the length of a specific limb (0 % normal limb, 50 % half of the limb length , 100 % limb amputation). Each color represents a different limb.

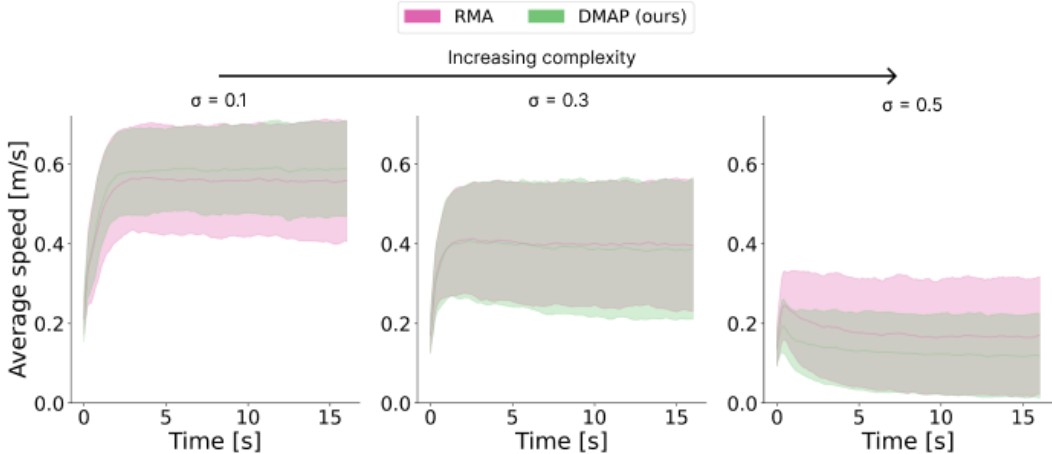

Figure F6: Average speed using the Ant agent across seeds and IID test body perturbations for different perturbation intensities. The adaptation speed to new morphological perturbation, i.e. time to reach asymptotic speed, is equivalent for RMA and DMAP.

where $x_t$ represents the trajectory at a specific time and $x_{t'}$ denotes the corresponding temporal derivative, and $\epsilon$ is a constant. We use this measure to study the the difference in trajectory tangling between attention and action/observation embeddings.

Here we quantified tangling for all four environments and IID test episodes. (Figure F7). Just like for the examples in the main text, we found that the attentional dynamics are less tangled than the inputs. This suggests that DMAP achieves control across morphological parameters by regularizing the dynamics.

To gain further insights into this mechanism, we verified if the attentional dynamics become less entangled during learning. Thus, we visualized how the attentional dynamics evolve throughout the reinforcement learning (Figure F8). While at an early stage the characteristic rotational dynamics are absent, it emerges as the training proceeds, progressively untangling the trajectories.

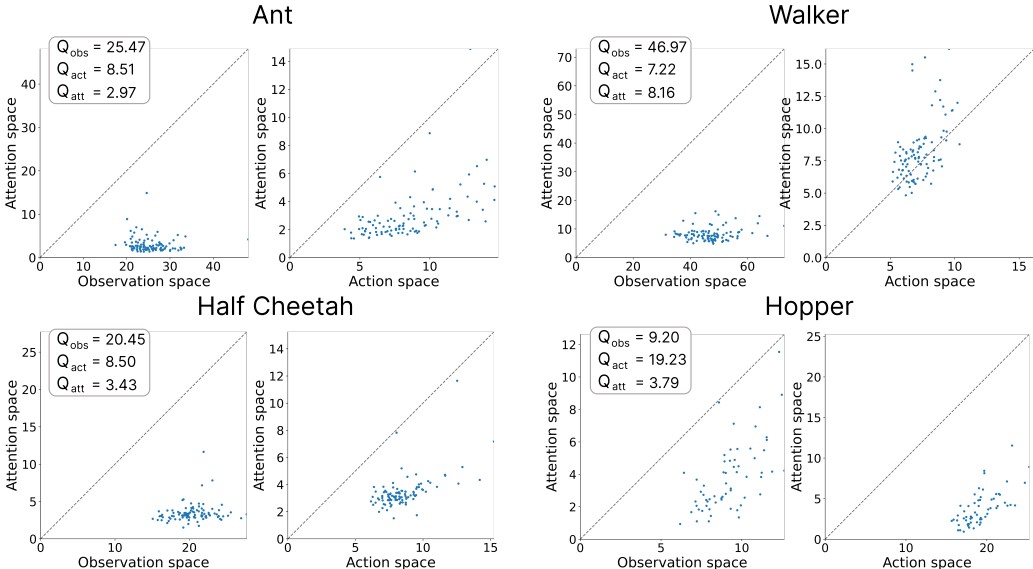

Figure F7: Scatter plot of the tangling measure for the attention embedding trajectories versus the observation and action ones. Each point represents the tangling measure for one of the 100 test body configurations averaged over time. The data refer to all the environments with $\sigma = 0.1$ and to the corresponding IID test body configurations. The large majority of the points lie below the threshold line, strongly suggesting that the attention trajectories are less tangled than those of the input actions and observations. This was also observed for larger perturbation intensities.

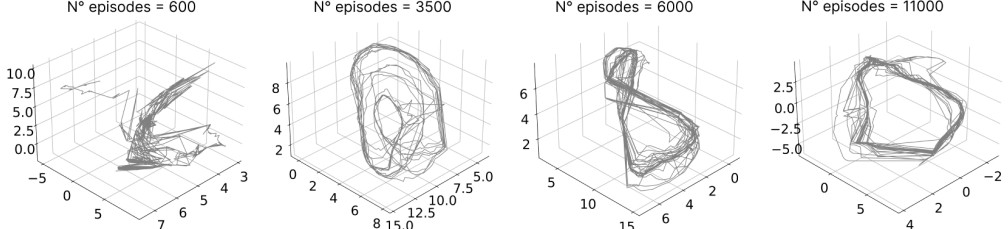

Figure F8: UMAP embedding of the attention matrix $K_t$ of a single training episode for different checkpoints during the training of an agent in the Ant environment with $\sigma = 0.1$. The training episode is indicated in the title. The characteristic rotational dynamics emerges as the agent learns how to walk.