# OpenReview forum: "DMAP: a Distributed Morphological Attention Policy for learning to locomote with a changing body"
_NeurIPS.cc/2022/Conference — NeurIPS 2022 Accept_

### Official Review · Reviewer_ZnNr · 2022-07-10

**Rating:** 7
**Confidence:** 4
**Soundness:** 3 good
**Presentation:** 2 fair
**Contribution:** 3 good

**Summary:**

This paper introduces DMAP, an algorithm that allows reinforcement learning to adapt the policy to locomotion environments where the character doesn’t have a fixed body. DMAP is based on an attention-based feature encoder.

**Questions:**

I think most of the questions are caused by the fact that too much information is squeezed into the main paper and details are skipped. Which makes me believe that it’s quite hard to reproduce from reading the paper alone. I would perhaps suggest to put some ablations in the appendix and expand on specifying the model itself.

- The fact that the character changes is equivalent with saying that the environment changes and as you mentioned this resembles the problem setting of zero-shot adaptation, or meta-learning. It would be interesting to see how methods and algorithms from the zero-shot adaptation literature trained in your environment settings compare to learning morphological changes with attention with respect to results or learning efficiency.

- In the distributed joint controllers, the proprioceptive state included in the state, is the local of just that specific joint or of the whole character?

- How are you producing the matrices K and V from the stack of independent representations? You call K “joint-channel attention”, is this produced via attention/transformer layers? I can’t find it specified in the description of the method.

- What do cells in V represent? Is the size V a hyperparameter?

- It’s hard to read figure 3 and 4 separately, I would suggest put the 4 lines together.

- Regarding the above, it seems, but I might be wrong, that the oracle is worse than DMAP. From my understanding the oracle should have full observability of the environment and should perform better. Do you have explanations for this?

- In section 5.4, I don’t understand how this ablation is performed. Are you removing $e_i$ from the input of each $\omega$ ? If so, doesn’t this become equivalent to the TCN architecture?

- Figure 6. Is there an explanation why the hip controller just looks at the front knee velocity and the knee controllers only the back knee velocity?

- Having $N_a$ separate policies will definitely increase computation time. How does learning speed compare to a single policy method? A way to ablate the importance of distributed joint controllers could be to give the full $E^{(t)}$ to a single policy that controls the full action.

- Regarding the point above, are separate policies feasible with a more complex character like humanoid?


**Limitations:**

Overall the method and the idea is really interesting. Most of my concerns are pointed out in the questions section.

I am willing to update my score after my concerns and questions have been clarified.

**Strengths And Weaknesses:**

- The idea behind this work is very interesting and the method shows good results.
- The paper motivates clearly the reasons of the architectural choices for DMAP pipeline.
- It ablates the different components to understand the effects of the choices.

---

> ### Author Response · Authors · 2022-08-02
> **Comparison to other zero-shot learning methods**
>
> In our analysis we focus on Rapid Motor Adaptation [R1] as well as the Oracle as a reference zero-shot adaptation algorithm because of its exceptional sample efficiency (in their experiments, only 0.5 seconds of transition history are proven sufficient for motor adaptation in difficult contexts). We find that DMAP can match its performance without the need of learning the morphology encoding from an Oracle agent, differently from the RMA policy trained end-to-end. Regarding the learning efficiency of DMAP, we bring to the attention of the reviewer the updated Figure 4 of the manuscript, in which we plot the learning curves of both the Oracle and DMAP. The number of episodes to reach convergence is very similar for both methods, meaning that DMAP is as efficient as a policy with full observability of the system state (proprioceptive information + morphological context).
>
> However, we do agree that comparing to meta-RL algorithms is interesting. In the short time between the notification of this review and the deadline for our response, we have tested the Garage implementation of PEARL, an off-policy meta-RL algorithm also used for motor control (https://garage.readthedocs.io/en/latest/_autoapi/garage/torch/algos/pearl/). We tested PEARL  on the perturbed Ant environment. We framed it as a meta-learning problem by defining random perturbations to use at training time, and testing the agent trained with PEARL on random test perturbations. Unfortunately, we have not succeeded at fully exploiting our computational resources with Garage, so the training is still at an early stage after a few days of training. The current results are: after 500k steps in the training environment, the Ant collects a cumulative reward of 434 in the sigma = 0.5 perturbation environment and 485 in the sigma = 0.3 perturbation environment. The current poor performance is likely due to too few iterations (and no hyperparameter tuning so far). We can include results for PEARL in the full manuscript, once we have more exhaustively evaluated it.
>
> [R1] Kumar, Ashish, et al. "RMA: Rapid motor adaptation for legged robots." arXiv preprint arXiv:2107.04034 (2021)
>
> [R2] Rakelly, Kate, Aurick Zhou, Deirdre Quiilen, Chelsea Finn, and Sergey Levine. 2019. “Efficient off-Policy Meta-Reinforcement Learning via Probabilistic Context Variables.” 36th International Conference on Machine Learning, ICML 2019 2019-June: 9291–9301.

---

> > ### Comment · Reviewer_ZnNr · 2022-08-09
> > **Thanks for the thorough answer.**
> >
> > Thanks to the authors for the detailed reply to each of my concerns. As I initially stated, I think this work has no flows and could be of high interest to the community. My main concern was related to the amount of information crammed in the original paper, and lack of some important details.
> > I have therefore updated the score of my review.

---

> ### Author Response · Authors · 2022-08-02
> **Technical questions about DMAP**
>
> *Question: In the distributed joint controllers, the proprioceptive state included in the state, is the local of just that specific joint or of the whole character?*
>
> For DMAP, we provide the complete proprioceptive state of all joints to each controller.
>
> For the revision, we have also done additional experiments, where only the proprioceptive state of the same joint is provided to the controller (local_DMAP). We find similar performance results (see below), suggesting that the attention mechanism is sufficient for achieving multi-joint coordination. This is interesting, as in biological systems, typically the sensors of one muscle do not directly (i.e., <= 1 synapse) modulate other muscles.
>
> Sigma:  0.1
> Reward:  2622  +-  13
>
> Sigma:  0.3
> Reward:  1690  +-  16
>
> Sigma:  0.5
> Reward:  934 +-  4
>
> *Q: How are you producing the matrices K and V from the stack of independent representations? You call K “joint-channel attention”, is this produced via attention/transformer layers? I can’t find it specified in the description of the method.*
>
> Each row of both the matrix K and the matrix V is a linear transformation of the single-channel output of the TCNs (i.e. the temporally filtered proprioceptive input). In other words, the network learns two weight matrices, one for K and one for V, shared across all input channels, to project the output of the TCNs into an adequately sized vector (number of actions for K, size of the morphology representation for V - detailed in the following answer). Before performing the product K^t V, a softmax operation is applied to the columns of K. We call K “joint-channel attention” because an element (i, j) of the matrix K represents the weight by which the i-th row of V (the encoding of the i-th joint state history) will be multiplied when generating the encoding vector e^(t)_j for the controller of the j-th action component.
> Thanks for raising this, we have clarified this point in the manuscript.
>
>
> *Q: What do cells in V represent? Is the size V a hyperparameter?*
>
> The matrix V of DMAP is analogous to the value matrix of a multi-head attention architecture. Each row of V contains an embedding resulting from the temporal filtering of one input channel (matrix H(t) in the architecture description, containing state and action history).
> The size of V is a hyperparameter, defined as “encoding size” in Table 4 in the appendix. For its value we drew inspiration from the RMA paper, where they used half as many latent variables for the environmental factor encoder as the number of environment parameters. Thus, in our case, for morphological perturbations of dimension 9 (Ant), we utilized encoding dimensions of size 4. We kept the same size for all the other architectures (Oracle, RMA, TCN, DMAP) and environments (Ant, Half Cheetah, Hopper, Walker) we tested.
>
> *Q: It’s hard to read figure 3 and 4 separately, I would suggest put the 4 lines together.*
>
> Thank you for this suggestion – in the revised manuscript, we have added the learning curve of the Oracle to figure 4, so that the reader can better compare it with DMAP.
>
> *Q: Regarding the above, it seems, but I might be wrong, that the oracle is worse than DMAP. From my understanding the oracle should have full observability of the environment and should perform better. Do you have explanations for this?*
>
> The performance for DMAP and the Oracle is indeed comparable – which is a key result, as DMAP does not have access to the morphology parameters.
> The two agents perform similarly in all four environments and three perturbation levels, except for the Walker, where the Oracle is stronger in the low perturbation regime (sigma 0.1) and DMAP is better in the high perturbation regime (sigma 0.5). The superior performance of DMAP in Walker sigma 0.5 might be due to the higher effectiveness of the attention-based encoder in that environment. We remark that the Oracle and DMAP do not only differ for the available information in input, but also in many architectural choices, which can explain why in certain tests DMAP is better. While the Oracle provides a strong reference performance, due to its access to the full environment context information, there is no guarantee that the reward it achieves is the maximum for any RL agent trained in that environment.
>
> *Q: Figure 6. Is there an explanation why the hip controller just looks at the front knee velocity and the knee controllers only the back knee velocity?*
>
> The attention map developed by DMAP varies for different seeds, indeed the gait patterns can vary across seeds. Furthermore, many state and action parameters are correlated during simple locomotion. This makes interpreting why specific connections emerge in the attention maps challenging. Despite this, across seeds, we often observe bilateral and back-front symmetry. In the future, we want to study this with more challenging locomotion tasks that decorrelate the states (e.g. non-flat terrains and also varying morphologies).

---

> ### Author Response · Authors · 2022-08-02
> **Ablations, hierarchical and distributed control**
>
> We thank the reviewer for the positive rating and clarifying questions. To clarify our contributions, we followed the suggestion by expanding the related works, model and experimental design description.
>
> *Q: In section 5.4, I don’t understand how this ablation is performed. Are you removing ei from the input of each ω ? If so, doesn’t this become equivalent to the TCN architecture?*
>
> We perform test-time ablation, i.e. we remove the encoding from the trained DMAP architecture without re-training the policy. In practice, we simply set it to 0. The purpose is to assess whether the agent actually “needs” the attention module’s input, or if the distributed policy networks are sufficient. Based on the results (showing a performance drop of 10-50% in appendix A.5), we conclude that the attention module is of fundamental importance, especially when the agent needs to dynamically balance itself (hopper and walker).
>
> The differences with the TCN architecture are the following:
> - The TCN encoder is different from the TCN component of dmap, as what we called TCN processes all the channels (joints) together (as in RMA), while DMAP separately
> - The TCN agent is allowed to use the encoding it creates with the history of proprioceptive states, while in our ablation study we are removing such encoding
>
> The network without the encodings is similar to the simple agent, but with a distributed controller instead of a centralized one. In order to extend the related work section and to clarify other sections of the paper, while adhering to the 9-page rule, we have moved this ablation experiment to the supplementary material (A.5).
>
> *Having separate policies will definitely increase computation time. How does learning speed compare to a single policy method? A way to ablate the importance of distributed joint controllers could be to give the full E(t) to a single policy that controls the full action.*
>
> Compared to the RMA network architecture, we reduced the network size in DMAP for the distributed controllers to a 2-layer fully-connected network, with 32 nodes per layer. This makes the
> - overall network size similar to the one used by RMA
> - training time was only marginally longer than that of RMA.
>
> Giving the whole E(t) to a single controller is an interesting suggestion (ablation). During the revision period, we did additional experiments for the Ant environment with a perturbation level of 0.5. We replaced the distributed controller with a single one, receiving the full E(t) caused a performance drop of around 5%. This further corroborates our conclusions about the importance of distributed control seen in biology.
>
> *Q: Regarding the point above, are separate policies feasible with a more complex character like humanoid?*
>
> As long as characters do not have too many degrees of freedom (DoF), DMAP is certainly applicable, but we have focussed on four agents so far. As such, DMAP is inspired by the sensorimotor system of animals, which can control highly complex dynamical systems with many DoF. For the agents we considered (Ant, Half Cheetah, Hopper, Walker) DMAP did efficiently locomote and we do hope that brain-inspired approaches like DMAP will allow us to tackle more complex characters in the future.

---

### Official Review · Reviewer_1n2P · 2022-07-10

**Rating:** 7
**Confidence:** 4
**Soundness:** 3 good
**Presentation:** 4 excellent
**Contribution:** 3 good

**Summary:**

- This paper studies the problem of zero shot adaptation in the context of changing bodies doing a continuous control task.

- Contrary to typical RL benchmarks that train to maximize expected rewards and test on the same metric, the metric here is to evaluate the adaptation capability of the learn policies under morphological changes.

- The key idea is to factor out each proprioceptive channel and learn local embeddings. An attention based architecture is then used to implement a form of dynamic gating to solve the control task.

- During training, the bodies are perturbed in simulation with a fixed distribution and then evaluated on test environment with unseen values.

- The key result is that this architecture is at par or even surpasses the performance of an oracle given unseen body configurations during test time.


**Questions:**

- It seems like Figure 2 varies the body w.r.t various parameters like size and shape. However, how does the approach fare under more radical transformations like broken legs? In order words, where does this approach fail?

- The qualitative analysis using UMAP and the observation stated "We speculate that this untangling mechanism might contribute
232 to the robustness of the policy to morphological perturbations." is quite interesting.

**Limitations:**

Discuss and perform evaluation of where the approach fails. Currently the morphological variations seems to be constrained to shape and size, but not breakage.

**Strengths And Weaknesses:**

Strengths:

- I really like the setting of this paper -- humans and animals have to learn under similar settings because the body changes everyday while the sensory motor loop still has to keep improving as the organism finds itself in new situations.

- This setup will also be increasingly relevant in the robotics community as we develop robots with more dexterity and human-like hands -- because as the embodiment becomes more complicated, the robot will needs lots of iterations and fixes due to wear-and-tear, and new methods will be needed to handle to adapt to these morphological changes.

- The experiments and writing are also clear. The approach is demonstrated on a set of continuous control tasks where the embodiment is structurally changed.

- Figure 6 shows emergent symmetries in the body space, which has natural biological analogs.

Weakness:

I wonder if the authors have a tried a sequence to sequence based approach (e.g Decision Transformer) to model this problem. This would be an "inductive-bias" free approach. This is less of a limitations in my opinion but more of a contrast to the bias that this paper proposes. I suspect this approach would be more data efficient or could adapt quicker with more drastic body amputations.

---

> ### Author Response · Authors · 2022-08-02
> **Limitations of DMAP and attention dynamic interpretation**
>
> We thank the reviewer for the constructive suggestions and the positive evaluation of our work.
>
> *Question: It seems like Figure 2 varies the body w.r.t various parameters like size and shape. However, how does the approach fare under more radical transformations like broken legs? In order words, where does this approach fail?*
>
> We have included an additional experiment in the supplementary to better assess limitations of DMAP (Section A. 7). Specifically, we quantify how the episode reward is affected when we progressively change the length of a limb in the ant (from 0 % perturbation intensity which is the normal limb to 100 % which corresponds to limb amputation). We find that the reward monotonically decreases similarly for different limbs with an increase in robustness, when DMAP is trained with higher perturbation intensity.
>
> *Question: The qualitative analysis using UMAP and the observation stated "We speculate that this untangling mechanism might contribute to the robustness of the policy to morphological perturbations." is quite interesting.*
>
> We agree with the reviewer that the untangling mechanism as a hypothesis for robust controllers is quite tantalizing. In other contexts, such as the analysis of neural population dynamics in the motor cortex, it has been suggested that untangled rotational-like dynamics might be a way to make the neural code robust to noise [R1]. In the future, we are planning to investigate the untangled dynamic of attention more in detail and its impact to the morphology perturbation robustness.
>
> [R1] Russo, Abigail A., et al. "Motor cortex embeds muscle-like commands in an untangled population response." Neuron 97.4 (2018): 953-966.

---

### Official Review · Reviewer_t2fY · 2022-07-11

**Rating:** 7
**Confidence:** 4
**Soundness:** 3 good
**Presentation:** 3 good
**Contribution:** 3 good

**Summary:**

Generalization in RL is a difficult problem. In this paper the authors investigate generalization to agents with changing morphology. This is accomplished be learning a policy that is conditioned on a history of observations and can use this information to learn a better joint attention over the parts of the state space the actuators on the robot should focus on when deciding actions. They evaluate this model against others on a task where the size and length of limbs change across a set of 4 agents. The results appear to be strong across prior methods evaluated in the paper.


---------

 I have updated my score after the initial responses.

I have updated my score further to reflect details covering other reviewers.

**Questions:**

Questions/Comments:

- It does not appear to be many experimental comparisons to prior work. For example, the experimental conditions described in the paper are very similar to the ones in Pearl, a meta-learning method that infers agent morphology to be able to do some tasks quickly. and that paper, they even perform this type of morphological modification to two different agent types as one of their motivational environments for being able to adjust for quicker adaptation.
  - Rakelly, Kate, Aurick Zhou, Deirdre Quiilen, Chelsea Finn, and Sergey Levine. 2019. “Efficient off-Policy Meta-Reinforcement Learning via Probabilistic Context Variables.” 36th International Conference on Machine Learning, ICML 2019 2019-June: 9291–9301.
- There is related work that fits this type of experimental analysis. If the claims in the paper and analysis where more clear it is likely the problem statement in the paper would align well with these prior works. It would also be important to compare to these prior works.
  - Nagabandi, Anusha, Ignasi Clavera, Simin Liu, Ronald S. Fearing, Pieter Abbeel, Sergey Levine, and Chelsea Finn. 2018. “Learning to Adapt in Dynamic, Real-World Environments Through Meta-Reinforcement Learning.” 7th International Conference on Learning Representations, ICLR 2019. arXiv. http://arxiv.org/abs/1803.11347v6.
  - Nagabandi, Anusha, Chelsea Finn, and Sergey Levine. 2019. “Deep Online Learning via Meta-Learning: Continual Adaptation for Model-Based RL.” 7th International Conference on Learning Representations, ICLR 2019, 1–15.
- It would be interesting to better understand the limitations of the attention module. Often these model require a large amount of data to train a useful representation. Some analysis to understand how many morphologies or tasks are needed to learn a generalizable representation would increase the strength of the work.
- Additional related work
  - Kurin, Vitaly, Maximilian Igl, Tim Rocktäschel, Wendelin Boehmer, and Shimon Whiteson. 2020. “My Body Is a Cage: The Role of Morphology in Graph-Based Incompatible Control.” arXiv [cs.LG]. arXiv. http://arxiv.org/abs/2010.01856.
  - Trabucco B, Phielipp M, Berseth G. AnyMorph: Learning Transferable Polices By Inferring Agent Morphology.  ICML 2022.

**Limitations:**

Some are discussed in the paper.

**Strengths And Weaknesses:**

Pros:
- Learning a more end-2-end approach for estimating and generalizing to these types of changes in morphology is easier to scale.
- The network design is sensible and should results in some interesting learned attention.
- The focus on the 3 main indicators of a robust sensory-motor system is interesting.

Cons:
- The particular motivation in the paper is somewhat confusing. Are there many scenarios where the length and thickness of agent limbs change frequently? More discussion of this motivation would help readers understand what larger problem this work is connected to.
- I also find the experiment section somewhat unclear. The design experiments are difficult to connect back to the claims in the paper. It would be very helpful to include some additional details on the particular types of experiments and why those particular experiments are being performed. For example, it's still somewhat unclear that the type of adaptation that is being claimed in the paper is mid-episode adaptation across a changing morphology. Or if the overall idea is to be able to train a policy that can adapt to other more apologies without additional episode data given. These details are important to understand what are the best methods or most related methods to compare in the experiment section.

---

> ### Author Response · Authors · 2022-08-02
> **Meta-RL comparison and related work**
>
> *Question: It does not appear to be many experimental comparisons to prior work. For example, the experimental conditions described in the paper are very similar to the ones in Pearl, a meta-learning method that infers agent morphology to be able to do some tasks quickly. and that paper, they even perform this type of morphological modification to two different agent types as one of their motivational environments for being able to adjust for quicker adaptation.*
>
> In our work we focus on Rapid Motor Adaptation (as well as the Oracle) as a reference zero-shot adaptation algorithm, because of its exceptional sample efficiency (in their experiments, only 0.5 seconds of transition history are sufficient for motor adaptation in difficult contexts). We find that DMAP can match its performance without the need of learning the morphology encoding from an Oracle agent.
> Additionally, we do agree that comparing to meta-RL algorithms is a great idea and we carefully reviewed the suggested works. We realized that the experimental setup of those previous studies [R1, R2, R3] do include perturbations causing a change in the transition dynamics of the environment (e.g. Walker-2D-Params and crippled leg). Such perturbations differ from our experimental setup as they do not change the agent morphology (although the visualization of crippled ant shows a leg of smaller size, in the actual code crippling corresponds to a perturbation of the actuation of the leg joint - https://github.com/iclavera/learning_to_adapt/blob/master/learning_to_adapt/envs/ant_env.py).
>
> Despite these differences, it is an excellent suggestion to also compare to PEARL [R1]. In the short time since the review appeared, we have tested the Garage [R4] implementation of PEARL on the perturbed Ant environment defined in our work. We framed it as a meta-learning problem by defining random perturbations to use at training time, and testing the agent trained with PEARL on random test perturbations. Unfortunately, we have not succeeded at fully exploiting our computational resources with Garage, so the training is still at an early stage after a few days of training. The current results are: after 500k steps in the training environment, the Ant collects a cumulative reward of 434 in the sigma = 0.5 perturbation environment and 485 in the sigma = 0.3 perturbation environment. The current poor performance is likely due to too few iterations (and no hyperparameter tuning so far). We can include results for PEARL in the full manuscript, once we have more exhaustively evaluated it.
>
> [R1] Rakelly et al. “Efficient off-Policy Meta-Reinforcement Learning via Probabilistic Context Variables.” 36th International Conference on Machine Learning, ICML 2019 2019-June: 9291–9301
> [R2] Nagabandi et al. “Learning to Adapt in Dynamic, Real-World Environments Through Meta-Reinforcement Learning.” 7th International Conference on Learning Representations, ICLR 2019. arXiv. http://arxiv.org/abs/1803.11347v6.
> [R3] Nagaband et al. “Deep Online Learning via Meta-Learning: Continual Adaptation for Model-Based RL.” 7th International Conference on Learning Representations, ICLR 2019, 1–15.
> [R4] Garage: A toolkit for reproducible reinforcement learning research, 2019, GitHub: https://github.com/rlworkgroup/garage
>
> *Question: There is related work that fits this type of experimental analysis. If the claims in the paper and analysis were more clear it is likely the problem statement in the paper would align well with these prior works. It would also be important to compare to these prior works.*
>
> We thank the reviewer for pointing us towards relevant work, including Amorpheus [R5] and the recently published AnyMorph [R6], addressing the complementary problem of controlling bodies in which even the connectivity graph of the body segments varies. We have included them in the related work section.
>
> [R5] Kurin, Vitaly, et al. "My body is a cage: the role of morphology in graph-based incompatible control." arXiv preprint arXiv:2010.01856 (2020).
> [R6] Trabucco B, Phielipp M, Berseth G. AnyMorph: Learning Transferable Polices By Inferring Agent Morphology. ICML 2022.

---

> > ### Comment · Reviewer_t2fY · 2022-08-05
> > **Response**
> >
> > Thank you as well for these comments on prior work and understanding.
> >
> > Generalization in RL is a fast-growing area, and it is challenging to keep up with the broad area of research. The additional analysis helps cover more of the small issues with the claims in the paper and the prior work. I have updated my score to reflect these updates. However, it would be good to understand how broadly this method can be used. Can it be used to solve the types of generalization studied across the additional papers that were cited? What are the limitations?

---

> ### Author Response · Authors · 2022-08-02
> **Biological motivation and attention module limitations**
>
> We thank the reviewer for the valuable feedback and questions. We tried to clarify the experimental section in the manuscript, specifying that morphological perturbations are always at the beginning of each episode, and that we evaluated the adaptation capability of the learned policies under morphological changes by considering rewards as a metric (without feedback). We hope that the manuscript will be clearer after those updates.
>
> *Q: The particular motivation in the paper is somewhat confusing. Are there many scenarios where the length and thickness of agent limbs change frequently? More discussion of this motivation would help readers understand what larger problem this work is connected to.*
>
> We do agree that changing the lengths and thickness on short timescales is a rather unusual perturbation. However, during development and also adulthood longer timescales (e.g., bodybuilding) those parameters can change substantially. Remarkably, animals and humans are robust to even abrupt changes in the length and weight of limb parameters. For instance, desert ants with elongated (“stilts”) or shortened legs (“stumps”) can immediately locomote, but interestingly misjudge traveled distances [R1]. Humans can also quickly walk with stilts [R2]. Moving with artificial weights is another short-timescale manipulation (related to width) to which animals and humans are very robust.
>
> [R1] Wittlinger, Matthias, Rüdiger Wehner, and Harald Wolf. "The ant odometer: stepping on stilts and stumps." science 312.5782 (2006): 1965-1967.
> [R2] Leurs, Françoise, et al. "Optimal walking speed following changes in limb geometry." Journal of Experimental Biology 214.13 (2011): 2276-2282
>
> *Question: It would be interesting to better understand the limitations of the attention module. Often these model require a large amount of data to train a useful representation. Some analysis to understand how many morphologies or tasks are needed to learn a generalizable representation would increase the strength of the work.*
>
> We understand the concerns of the reviewer about the amount of data which might be necessary to learn meaningful representations through the attention module, as these architectures are normally data-eager. In the context of our experiments, we might encounter two different data bottlenecks: the number of interactions between the agent and the environment to reach a high average episode reward and the number of morphologies that need to be observed during the training to generalize to unseen ones at test time. Regarding the number of interactions with the environment, in the updated figure 4 of the manuscript we can observe that the learning curves of Oracle and DMAP are very similar for most agents, meaning that the more complex network architecture of DMAP does not introduce any sample inefficiency. Regarding the number of different morphologies to be observed during the training, we ran additional experiments in the Ant environment, limiting the number of such morphologies. We trained DMAP sampling uniformly at random, in each episode, one array of body parameters first from a set of 10 and then from a set of 100 (in contrast to the standard setup of our work, in which each episode starts with an unseen morphology), sampled from sigma=0.5 perturbation intensity. Here we report the average (+- SEM) results obtained across 3 random seeds, as a function of the number of morphologies observed during the training:
>
> - 10 perturbations: 693 +- 37 (3 seeds)
> - 100 perturbations: 859 +- 6 (3 seeds)
> - New perturbation at each episode: 960 +-14 (standard experimental condition from the paper, 5 seeds)
>
> This suggests that collecting experience with a wide range of perturbations during the training is critical for DMAP to achieve high performance. However, with as little as 10 morphologies observed during training, in this specific environment configuration the agent could achieve approx. 70% of the performance of the agent trained with the standard setup of our other experiments. This is an over 90% performance improvement over the simple agent in the same perturbation level, which instead observes one new body configuration at each episode. With 100 perturbations, instead, DMAP reaches approx. 90% of its full performance.

---

> > ### Comment · Reviewer_t2fY · 2022-08-05
> > **Comments Appreciated**
> >
> > Thank you for your thoughtful and clear comments.
> >
> > ** Type of generalization to changes in morphology**
> > - It is a difficult line between the different types of generalization being studied across many types of work. For example, what is the meaningful difference between an agent that changes size and thickness of limbs vs adding or subtracting one is non obvious. However, prior methods may not apply to this case as easily.
> >
> > **Attention Module limitation**
> > - The additional analysis is appreciated. Understanding the lower limit of morphologies needed for training will benefit practitioners applying this method to more physical systems.

---

### Official Review · Reviewer_WjH3 · 2022-07-17

**Rating:** 7
**Confidence:** 4
**Soundness:** 3 good
**Presentation:** 3 good
**Contribution:** 3 good

**Summary:**

The authors tackle the problem of learning to locomote with morphological perturbations, e.g. when the length and the thickness of different body parts vary. They propose a biologically-inspired, attention-based policy network architecture. It combines a distributed policy, with individual controllers for each joint, and an attention mechanism, to dynamically gate sensory information from different body parts.

They ablate against two baselines: 1) A policy based on the proprioceptive state, that performs poorly with highly variable body configurations, and 2) an (oracle) agent with access to a learned encoding of the perturbation that performs significantly better. (And RMA and TCN)


**Questions:**

-

**Limitations:**

Yes

**Strengths And Weaknesses:**

Strengths
- The paper is written well and is easy to follow
- The challenge of dealing with unknown morphological perturbations is interesting and relevant.
- The application of RMA as a form of system identification is very helpful to compare the approach to inferred information (sysid) in addition to a baseline that uses ground truth information (Oracle)
- DMAP convincingly outperforms all baselines
- The visuals are great. Especially Figures 6 and 7 give an interesting intuition

Weaknesses
- The presented architecture seems similar to common approaches in hierarchical RL. A comparison seems appropriate. E.g. [R1] but I can imagine that there are quite a few approaches out there
- Literature on hierarchical RL would be helpful
- All methods were trained with SAC. It is hard to estimate the impact of entropy regularization on the proposed method. It would be helpful to also include DDPG/TD3
- RMA was able to identify the morphological context in 30 transitions (0.5s). It seems like the morphological changes in the presented tasks are easy to infer. One could say that the task itself is not very hard and so the benefit of the presented approach is in question. It might be helpful to develop even harder environments up to the point where RMA fails or takes significant amount of time. How well would the presented algorithm fare?


[R1] Rao, D., Sadeghi, F., Hasenclever, L., Wulfmeier, M., Zambelli, M., Vezzani, G., ... & Hadsell, R. (2021). Learning transferable motor skills with hierarchical latent mixture policies. arXiv preprint arXiv:2112.05062.

---

> ### Author Response · Authors · 2022-08-02
> **Expansion of related work, SAC entropy and RMA comparison**
>
> We thank the reviewer for the positive evaluation and for the valuable suggestions. We have extended the related work section to include hierarchical RL as previous studies in the field have investigated network architectures to promote adaptation in motor control (e.g., [R1] as mentioned by the reviewer, but also [R2, R3]).
>
> *Q: All methods were trained with SAC. It is hard to estimate the impact of entropy regularization on the proposed method. It would be helpful to also include DDPG/TD3.*
>
> We choose the Soft Actor-Critic (SAC) algorithm for all our experiments, because of its state-of-the-art performance for unperturbed environments (the agents trained with SAC in [R4] are the current leaders in average reward for Ant, HalfCheetah, Hopper and Walker in the PyBullet simulator, according to this leaderboard: https://paperswithcode.com/task/continuous-control).
>
> We agree that the entropy regularization of the algorithm could provide a relevant contribution to make the algorithm more robust in out-of-distribution (OOD) testing.
> However, a previous study on the OOD performance of SAC and TD3 in locomotion environments did not show a clear difference between the two algorithms [R5]. The same work also studies the relevance of the entropy coefficient, without finding a correlation between higher regularization and better OOD performance.
> Importantly, all algorithms in our manuscript (simple, oracle, RMA, DMAP) were trained with SAC, so the comparison is also fair. For this purpose, we also adapted the original Rapid Motor Adaptation (RMA) [R6] algorithm, which was originally based on Proximal Policy Optimization (PPO), to use SAC.
>
> *Q: RMA was able to identify the morphological context in 30 transitions (0.5s). It seems like the morphological changes in the presented tasks are easy to infer. One could say that the task itself is not very hard and so the benefit of the presented approach is in question. It might be helpful to develop even harder environments up to the point where RMA fails or takes significant amount of time. How well would the presented algorithm fare?*
>
> The choice of providing 30 transitions to RMA to perform context identification is based on the parameter choice outlined in the original paper [R6], where the authors show that the same time interval is sufficient for the quadruped robot to adapt to a variety of environment perturbations. This rather remarkable result motivated our experimental hypothesis that the same short transition history might be enough to infer the morphological context. Our experiments confirmed this.
>
> We could not perform online changes of the morphological parameters in PyBullet within an episode, therefore we compared speeds of RMA and DMAP when starting to run from resting position with an unseen morphology. We performed this analysis across seeds and test episodes using the Ant agent (Appendix section A.6). We find that for all perturbation intensities DMAP and RMA reach the final speed in about the same time.
>
> [R1] Rao, D., Sadeghi, F., Hasenclever, L., Wulfmeier, M., Zambelli, M., Vezzani, G., ... & Hadsell, R. (2021). Learning transferable motor skills with hierarchical latent mixture policies. arXiv preprint arXiv:2112.05062.
>
> [R2] Alexander C Li, Carlos Florensa, Ignasi Clavera, and Pieter Abbeel. Sub-policy adaptation for hierarchical reinforcement learning. arXiv preprint arXiv:1906.05862, 2019
>
> [R3] Andrew Levy, Robert Platt, and Kate Saenko. Hierarchical actor-critic. arXiv preprint arXiv:1712.00948, 12, 2017
>
> [R4] Raffin, Antonin, Jens Kober, and Freek Stulp. "Smooth exploration for robotic reinforcement learning." Conference on Robot Learning. PMLR, 2022
>
> [R5] Mann, Khushdeep Singh, et al. "Out-of-distribution generalization of internal models is correlated with reward." Self-Supervision for Reinforcement Learning Workshop-ICLR 2021. 2021.
>
> [R6] Kumar, Ashish, et al. "Rma: Rapid motor adaptation for legged robots." arXiv preprint arXiv:2107.04034 (2021)

---

### Meta-Review · Area_Chair_DcPx · 2022-08-26

**Recommendation:** Accept
**Confidence:** Certain

**Metareview:**

This paper examines the problem of learning locomotion for a simulated robot body, when the length and thickness of the body parts are varied.  The proposed method generates a distributed policy with controllers for each joint, and an attention mechanism that dynamically gates sensory information from different body parts.  The method performed well when compared to a policy based on proprioceptive state, and performs comparably to an oracle method given an embedding of the perturbed parameters.

Strengths of the paper noted by the reviewers included the clear writing, the relevance of the problem, and the convincing results.  The author response addressed potential limitations of the work raised by each of the reviewers. The reviewers were satisfied with the clarifications provided by the authors, and raised no additional concerns.

Four reviewers indicate to accept the paper for its clear contributions on how to learn locomotion policies when the body parameters are changed.  The paper is therefore accepted.

**Award:**

No

---

### Decision · Program_Chairs · 2022-09-14

Accept